# Look More but Care Less in Video Recognition

**Yitian Zhang**[1]* **Yue Bai**[1] **Huan Wang**[1] **Yi Xu**[1] **Yun Fu**[1,2]
[1]Department of Electrical and Computer Engineering, Northeastern University
[2]Khoury College of Computer Science, Northeastern University

## Abstract

Existing action recognition methods typically sample a few frames to represent each video to avoid the enormous computation, which often limits the recognition performance. To tackle this problem, we propose Ample and Focal Network (AFNet), which is composed of two branches to *utilize more frames but with less computation*. Specifically, the Ample Branch takes all input frames to obtain abundant information with condensed computation and provides the guidance for Focal Branch by the proposed Navigation Module; the Focal Branch squeezes the temporal size to only focus on the salient frames at each convolution block; in the end, the results of two branches are adaptively fused to prevent the loss of information. With this design, we can introduce more frames to the network but cost less computation. Besides, we demonstrate AFNet can *utilize fewer frames while achieving higher accuracy* as the dynamic selection in intermediate features enforces implicit temporal modeling. Further, we show that our method can be extended to reduce spatial redundancy with even less cost. Extensive experiments on five datasets demonstrate the effectiveness and efficiency of our method. Our code is available at https://github.com/BeSpontaneous/AFNet-pytorch.

## 1 Introduction

Online videos have grown wildly in recent years and video analysis is necessary for many applications such as recommendation [6], surveillance [4, 5] and autonomous driving [31, 17]. These applications require not only accurate but also efficient video understanding algorithms. With the introduction of deep learning networks [3] in video recognition, there has been rapid advancement in the performance of the methods in this area. Though successful, these deep learning methods often cost huge computation, making them hard to be deployed in the real world.

In video recognition, we need to sample multiple frames to represent each video which makes the computational cost scale proportionally to the number of sampled frames. In most cases, a small proportion of all the frames is sampled for each input, which only contains limited information of the original video. A straightforward solution is to sample more frames to the network but the computation expands proportionally to the number of sampled frames.

There are some works proposed recently to dynamically sample salient frames [29, 16] for higher efficiency. The selection step of these methods is made before the frames are sent to the classification network, which means the information of those unimportant frames is totally lost and it consumes a considerable time for the selection procedure. Some other methods proposed to address the spatial redundancy in action recognition by adaptively resizing the resolution based on the importance of each frame [23], or cropping the most salient patch for every frame [28]. However, these methods still completely abandon the information that the network recognizes as unimportant and introduce a policy network to make decisions for each sample which leads to extra computation and complicates the training strategies.

---

*Corresponding Author: markcheung9248@gmail.com.

36th Conference on Neural Information Processing Systems (NeurIPS 2022).

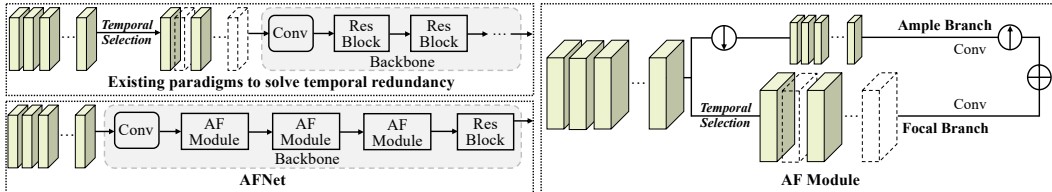

Figure 1: Comparisons between existing methods and our proposed Ample and Focal Network (AFNet). Most existing works reduce the redundancy in data at the beginning of the deep networks which leads to the loss of information. We propose a two-branch design which processes frames with different computational resources within the network and preserves all input information as well.

In our work, we go from another perspective compared with previous works. We propose a method which makes frame selection within the classification network. Shown in Figure 1, we design an architecture called Ample and Focal Network (AFNet) which is composed of two branches: the ample branch takes a glimpse of all the input features with lightweight computation as we downsample the features for smaller resolution and further reduce the channel size; the focal branch receives the guidance from the proposed navigation module to squeeze the temporal size by only computing on the selected frames to save cost; in the end, we adaptively fuse the features of these two branches to prevent the information loss of the unselected frames.

In this manner, the two branches are both very lightweight and we enable AFNet to look broadly by sampling more frames and stay focused on the important information for less computation. Considering these two branches in a uniform manner, on the one hand, we can avoid the loss of information compared to other dynamic methods as the ample branch preserves the information of all the input; on the other hand, we can restrain the noise from the unimportant frames by deactivating them in each convolutional block. Further, we have demonstrated that the dynamic selection strategy at intermediate features is beneficial for temporal modeling as it implicitly implements frame-wise attention which can enable our network to utilize fewer frames while obtaining higher accuracy. In addition, instead of introducing a policy network to select frames, we design a lightweight navigation module which can be plugged into the network so that our method can easily be trained in an end-to-end fashion. Furthermore, AFNet is compatible with spatial adaptive works which can help to further reduce the computations of our method.

We summarize the main contributions as follows:

- We propose an adaptive two-branch framework which enables 2D-CNNs to *process more frames with less computational cost*. With this design, we not only prevent the loss of information but strengthen the representation of essential frames.
- We propose a lightweight navigation module to *dynamically select salient frames* at each convolution block which can easily be trained in an *end-to-end* fashion.
- The selection strategy at intermediate features not only empowers the model with strong flexibility as different frames will be selected at different layers, but also enforces *implicit temporal modeling* which enables AFNet to *obtain higher accuracy with fewer frames*.
- We have conducted comprehensive experiments on five video recognition datasets. The results show the *superiority of AFNet compared to other competitive methods*.

## 2 Related Work

### 2.1 Video Recognition

The development of deep learning in recent years serves as a huge boost to the research of video recognition. A straightforward method for this task is using 2D-CNNs to extract the features of sampled frames and use specific aggregation methods to model the temporal relationships across frames. For instance, TSN [27] proposes to average the temporal information between frames. While TSM [20] shifts channels with adjacent frames to allow information exchange at temporal dimension. Another approach is to build 3D-CNNs to for spatiotemporal learning, such as C3D [26], I3D [3] and SlowFast [8]. Though being shown effective, methods based on 3D-CNNs are computationally expensive, which brings great difficulty in real-world deployment.

While the two-branch design has been explored by SlowFast, our motivation and detailed structure are different from it in the following ways: 1) network category: SlowFast is a static 3D model, but

AFNet is a dynamic 2D network; 2) motivation: SlowFast aims to collect semantic information and changing motion with branches at different temporal speeds for better performance, while AFNet is aimed to dynamically skip frames to save computation and the design of two-branch structure is to prevent the information loss; 3) specific design: AFNet is designed to downsample features for efficiency at ample branch while SlowFast processes features in the original resolution; 4) temporal modeling: SlowFast applies 3D convolutions for temporal modeling, AFNet is a 2D model which is enforced with implicit temporal modeling by the designed navigation module.

## 2.2 Redundancy in Data

The efficiency of 2D-CNNs has been broadly studied in recent years. While some of the works aim at designing efficient network structure [13], there is another line of research focusing on reducing the intrinsic redundancy in image-based data [32, 11]. In video recognition, people usually sample limited number of frames to represent each video to prevent numerous computational costs. Even though, the computation for video recognition is still a heavy burden for researchers and a common strategy to address this problem is reducing the temporal redundancy in videos as not all frames are essential to the final prediction. [33] proposes to use reinforcement learning to skip frames for action detection. There are other works [29, 16] dynamically sampling salient frames to save computational cost. As spatial redundancy widely exists in image-based data, [23] adaptively processes frames with different resolutions. [28] provides the solution as cropping the most salient patch for each frame. However, the unselected regions or frames of these works are completely abandoned. Hence, there will be some information lost in their designed procedures. Moreover, most of these works adopt a policy network to make dynamic decisions, which introduces additional computation somehow and splits the training into several stages. In contrast, our method adopts a two-branch design, allocating different computational resources based on the importance of each frame and preventing the loss of information. Besides, we design a lightweight navigation module to guide the network where to look, which can be incorporated into the backbone network and trained in an end-to-end way. Moreover, we validate that the dynamic frame selection at intermediate features will not only empower the model with strong flexibility as different frames will be selected at different layers, but result in learned frame-wise weights which enforce implicit temporal modeling.

## 3 Methodology

Intuitively, considering more frames enhances the temporal modeling but results in higher computational cost. To efficiently achieve the competitive performance, we propose AFNet to involve more frames but wisely extract information from them to keep the low computational cost. Specifically, we design a two-branch structure to treat frames differently based on their importance and process the features in an adaptive manner which can provide our method with strong flexibility. Besides, we demonstrate that the dynamic selection of frames in the intermediate features results in learned frame-wise weights which can be regarded as implicit temporal modeling.

### 3.1 Architecture Design

As is shown in Figure 2, we design our Ample and Focal (AF) module as a two-branch structure: the ample branch (top) processes abundant features of all the frames in a lower resolution and a squeezed channel size; the focal branch (bottom) receives the guidance from ample branch generated by the navigation module and makes computation only on the selected frames. Such design can be conveniently applied to existing CNN structures to build AF module.

**Ample Branch.** The ample branch is designed to involve all frames with cheap computation, which serves as 1) guidance to select salient frames to help focal branch to concentrate on important information; 2) a complementary stream with focal branch to prevent the information loss via a carefully designed fusion strategy.

Formally, we denote video sample $i$ as $v^i$, containing $T$ frames as $v^i = \left\{ f_1^i, f_2^i, ..., f_T^i \right\}$. For convenience, we omit the superscript $i$ in the following sections if no confusion arises. We denote the input of ample branch as $v_x \in \mathbb{R}^{T \times C \times H \times W}$, where $C$ represents the channel size and $H \times W$ is the spatial size. The features generated by the ample branch can be written as:

$$v_{y^a} = F^a \left( v_x \right), \tag{1}$$

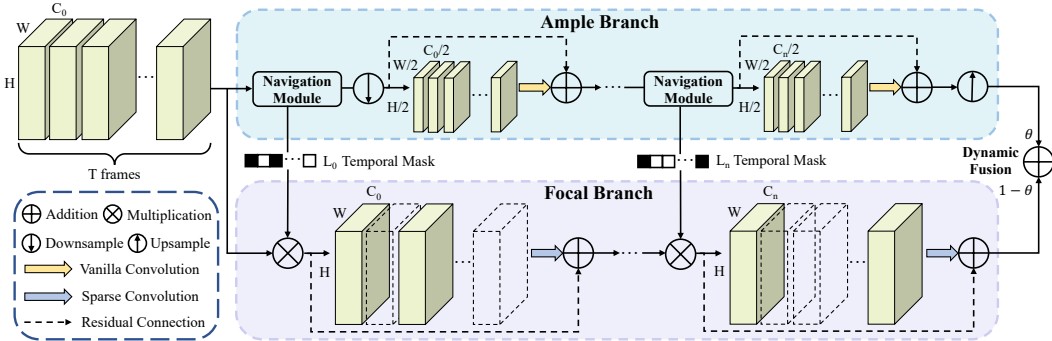

Figure 2: Architecture of AF module. The module is composed of two branches, the ample branch would process all the input features in a lower resolution and reduced channel size; while the focal branch would only compute the features of salient frames (colored features) guided by our proposed navigation module. The results of two branches are adaptively fused at the end of AF module so that we can prevent the loss of information.

where $v_{y^a} \in \mathbb{R}^{T \times (C_o/2) \times (H_o/2) \times (W_o/2)}$ represents the output of ample branch and $F^a$ stands for a series of convolution blocks. While the channel, height, width at focal branch are denoted as $C_o$, $H_o$, $W_o$ correspondingly. We set the stride of the first convolution block to 2 to downsample the resolution of this branch and we upsample the feature at the end of this branch by nearest interpolation.

**Navigation Module.** The proposed navigation module is designed to guide the focal branch where to look by adaptively selecting the most salient frames for video $v^i$.

Specifically, the navigation module generates a binary temporal mask $L_n$ using the output from the $n$-th convolution block in ample branch $v_{y^a_n}$. At first, average pooling is applied to $v_{y^a_n}$ to resize the spatial dimension to $1 \times 1$, then we perform convolution to transform the channel size to 2:

$$\tilde{v}_{y^a_n} = \text{ReLU}\left(\text{BN}\left(W_1 * \text{Pool}\left(v_{y^a_n}\right)\right)\right), \tag{2}$$

where $*$ stands for convolution and $W_1$ denotes the weights of the $1 \times 1$ convolution. After that, we reshape the dimension of feature $\tilde{v}_{y^a_n}$ from $T \times 2 \times 1 \times 1$ to $1 \times (2 \times T) \times 1 \times 1$ so that we can model the temporal relations for each video from channel dimension by:

$$p^t_n = W_2 * \tilde{v}_{y^a_n}, \tag{3}$$

where $W_2$ represents the weights of the second $1 \times 1$ convolution and it will generate a binary logit $p^t_n \in \mathbb{R}^2$ for each frame $t$ which denotes whether to select it.

However, directly sampling from such discrete distribution is non-differentiable. In this work, we apply Gumbel-Softmax [14] to resolve this non-differentiability. Specifically, we generate a normalized categorical distribution by using Softmax:

$$\pi = \left\{ l_j \mid l_j = \frac{\exp\left(p^{t_j}_n\right)}{\exp\left(p^{t_0}_n\right) + \exp\left(p^{t_1}_n\right)} \right\}, \tag{4}$$

and we draw discrete samples from the distribution $\pi$ as:

$$L = \arg\max_j \left(\log l_j + G_j\right), \tag{5}$$

where $G_j = -\log(-\log U_j)$ is sampled from a Gumbel distribution and $U_j$ is sampled from Unif(0,1) which is a uniform distribution. As $\arg\max$ cannot be differentiated, we relax the discrete sample $L$ in backpropagation via Softmax:

$$\hat{l}_j = \frac{\exp\left(\left(\log l_j + G_j\right)/\tau\right)}{\sum_{k=1}^{2} \exp\left(\left(\log l_k + G_k\right)/\tau\right)}, \tag{6}$$

the distribution $\hat{l}$ will become a one-hot vector when the temperature factor $\tau \to 0$ and we let $\tau$ decrease from 1 to 0.01 during training.

**Focal Branch.** The focal branch is guided by the navigation module to only compute the selected frames, which diminishes the computational cost and potential noise from redundant frames.

The features at the $n$-th convolution block in this branch can be denoted as $v_{y_n^f} \in \mathbb{R}^{T \times C_o \times H_o \times W_o}$. Based on the temporal mask $L_n$ generated from the navigation module, we select frames which have corresponding non-zero values in the binary mask for each video and apply convolutional operations only on these extracted frames $v'_{y_n^f} \in \mathbb{R}^{T_l \times C_o \times H_o \times W_o}$:

$$v'_{y_n^f} = F_n^f \left( v'_{y_{n-1}^f} \right), \tag{7}$$

where $F_n^f$ is the $n$-th convolution blocks at this branch and we set the group number of convolutions to 2 in order to further reduce the computations. After the convolution operation at $n$-th block, we generate a zero-tensor which shares the same shape with $v_{y_n^f}$ and fill the value by adding $v'_{y_n^f}$ and $v_{y_{n-1}^f}$ with the residual design following [12].

At the end of these two branches, inspired by [1, 11], we generate a weighting factor $\theta$ by pooling and linear layers to fuse the features from two branches:

$$v_y = \theta \odot v_{y^a} + (1 - \theta) \odot v_{y^f}, \tag{8}$$

where $\odot$ denotes the channel-wise multiplication.

## 3.2 Implicit Temporal Modeling

While our work is mainly designed to reduce the computation in video recognition like [28, 24], we demonstrate that AFNet enforces implicit temporal modeling by the dynamic selection of frames in the intermediate features. Considering a TSN[27] network which adapts vanilla ResNet[12] structure, the feature at the $n$-th convolutional block in each stage can be written as $v_n \in \mathbb{R}^{T \times C \times H \times W}$. Thus, the feature at $n + 1$-th block can be represented as:

$$\begin{aligned} v_{n+1} &= v_n + F_{n+1} (v_n) \\ &= (1 + \Delta v_{n+1}) \, v_n, \end{aligned} \tag{9}$$

$$\Delta v_{n+1} = \frac{F_{n+1} (v_n)}{v_n}, \tag{10}$$

where $F_{n+1}$ is the $n + 1$-th convolutional block and we define $\Delta v_{n+1}$ as the coefficient learned from this block. By that we can write the output of this stage $v_N$ as:

$$v_N = \left[ \prod_{n=2}^{N} (1 + \Delta v_n) \right] * v_1. \tag{11}$$

Similarly, we define the features in ample and focal branch as:

$$v_{y_N^a} = \left[ \prod_{n=2}^{N} \left( 1 + \Delta v_{y_n^a} \right) \right] * v_{y_1}, \tag{12}$$

$$v_{y_N^f} = \left[ \prod_{n=2}^{N} \left( 1 + L_n * \Delta v_{y_n^f} \right) \right] * v_{y_1}, \tag{13}$$

where $L_n$ is the binary temporal mask generated by Equation 5 and $v_{y_1}$ denotes the input of this stage. Based on Equation 8, we can get the output of this stage as:

$$\begin{aligned} v_{y_N} &= \theta \odot v_{y_N^a} + (1 - \theta) \odot v_{y_N^f} \\ &= \left\{ \theta \odot \left[ \prod_{n=2}^{N} \left( 1 + \Delta v_{y_n^a} \right) \right] + (1 - \theta) \odot \left[ \prod_{n=2}^{N} \left( 1 + L_n * \Delta v_{y_n^f} \right) \right] \right\} * v_{y_1}. \end{aligned} \tag{14}$$

As $L_n$ is a temporal-wise binary mask, it will decide whether the coefficient $\Delta v_{y_n^f}$ will be calculated in each frame at every convolutional block. Considering the whole stage is made up of multiple convolutional blocks, the series multiplication of focal branch's output with the binary mask $L_n$ will approximate soft weights. This results in learned frame-wise weights in each video which we regard as implicit temporal modeling. Although we do not explicitly build any temporal modeling module, the generation of $L_n$ in Equation 3 has already taken the temporal information into account so that the learned temporal weights equal performing implicit temporal modeling at each stage.

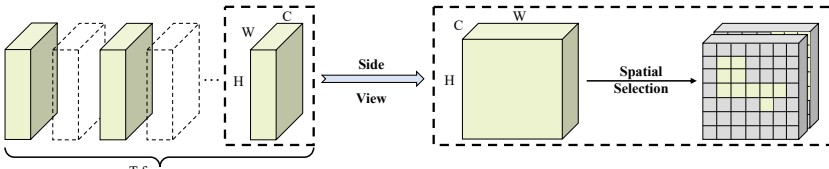

Figure 3: Illustration of extending AFNet to reduce spatial redundancy to further improve the efficiency. Only the colored area will be calculated at the inference stage.

### 3.3 Spatial Redundancy Reduction

In this part, we show that our approach is compatible with methods that aim to solve the problem of spatial redundancy. We extend the navigation module by applying similar procedures with the temporal mask generation and the work [11] to generate a spatial logit for the $n$-th convolution block which is shown in Figure 3:

$$q_n^t = W_4 * \left( \text{Pool} \left( \text{ReLU} \left( \text{BN} \left( W_3 * v_{y_n^a} \right) \right) \right) \right), \quad (15)$$

where $W_3$ denotes the weights of the $3 \times 3$ convolution and $W_4$ stands for the weights of convolution with kernel size $1 \times 1$. After that, we still use Gumbel-Softmax to sample from discrete distribution to generate spatial mask $M_n$ and navigate the focal branch to merely focus on the salient regions of the selected frames to further reduce the cost.

### 3.4 Loss functions

Inspired by [27], we take the average of each frame's prediction to represent the final output of the corresponding video and our optimization objective is minimizing:

$$\mathcal{L} = \sum_{(v,y)} \left[ -y \log \left( P\left(v\right) \right) + \lambda \cdot \sum_{n=1}^{N} \left( r - RT \right)^2 \right]. \quad (16)$$

The first term is the cross-entropy between predictions $P\left(v\right)$ for input video $v$ and the corresponding one-hot label $y$. We denote $r$ in the second term as the ratio of selected frames in every mini-batch and $RT$ as the target ratio which is set before the training ($RS$ is the target ratio when extending navigation module to reduce spatial redundancy). We let $r$ approximate $RT$ by adding the second loss term and manage the trade-off between efficiency and accuracy by introducing a factor $\lambda$ which balances these two terms.

## 4 Empirical Validation

In this section, we conduct comprehensive experiments to validate the proposed method. We first compare our method with plain 2D CNNs to demonstrate that our AF module implicitly implements temporal-wise attention which is beneficial for temporal modeling. Then, we validate AFNet's efficiency by introducing more frames but costing less computation compared with other methods. Further, we show AFNet's strong performance compared with other efficient action recognition frameworks. Finally, we provide qualitative analysis and extensive ablation results to demonstrate the effectiveness of the proposed navigation module and two-branch design.

**Datasets.** Our method is evaluated on five video recognition datasets: (1) Mini-Kinetics [23, 24] is a subset of Kinetics [15] which selects 200 classes from Kinetics, containing 121k training videos and 10k validation videos; (2) ActivityNet-v1.3 [2] is an untrimmed dataset with 200 action categories and average duration of 117 seconds. It contains 10,024 video samples for training and 4,926 for validation; (3) Jester is a hand gesture recognition dataset introduced by [22]. The dataset consists of 27 classes, with 119k training videos and 15k validation videos; (4) Something-Something V1&V2 [10] are two human action datasets with strong temporal information, including 98k and 194k videos for training and validation respectively.

**Data pre-processing.** We sample 8 frames uniformly to represent every video on Jester, Mini-Kinetics, and 12 frames on ActivityNet and Something-Something to compare with existing works unless specified. During training, the training data is randomly cropped to $224 \times 224$ following [35], and we perform random flipping except for Something-Something. At inference stage, all frames are center-cropped to $224 \times 224$ and we use one-crop one-clip per video for efficiency.

**Implementation details.** Our method is bulit on ResNet50 [12] in default and we replace the first three stages of the network with our proposed AF module. We first train our two-branch network from scratch on ImageNet for fair comparisons with other methods. Then we add the proposed navigation module and train it along with the backbone network on video recognition datasets. In our implementations, RT denotes the ratio of selected frames while RS represents the ratio of select regions which will decrease from 1 to the number we set before training by steps. We let the temperature $\tau$ in navigation module decay from 1 to 0.01 exponentially during training. Due to limited space, we include more details of implementation in supplementary material.

### 4.1 Comparisons with Existing Methods

**Less is more.** At first, we implement AFNet on Something-Something V1 and Jester datasets with 8 sampled frames. We compare it with the baseline method TSN as both methods do not explicitly build temporal modeling module and are built on ResNet50. In Table 1, our method AFNet(RT=1.00) shows similar performance with TSN when selecting all the frames. Nonetheless, when we select fewer frames in AFNet, it exhibits much higher accuracy compared to TSN and AFNet(RT=1.00) which achieves *Less is*

Table 1: Comparisons with baseline method on Something-Something V1 and Jester datasets.

| Method | Frame | Sth-Sth V1 | Jester |
|---|---|---|---|
| | | Top-1 Acc. | Top-1 Acc. |
| TSN [27](our imp) | 8 | 18.6% | 83.5% |
| AFNet (RT=1.00) | 8 | 19.2% | 83.6% |
| AFNet (RT=0.50) | 8 | **26.8**% | **89.2**% |
| AFNet (RT=0.25) | 8 | **27.7**% | **89.2**% |
| AFNet (soft-weights) | 8 | **27.0**% | **89.9**% |

*More* by utilizing less frames but obtaining higher accuracy. The results may seem counterintuitive as seeing more frames is usually beneficial for video recognition. The explanation is that the two-branch design of AFNet can preserve the information of all input frames and the selection of salient frames at intermediate features implements implicit temporal modeling as we have analyzed in Section 3.2. As the binary mask learned by the navigation module will decide whether the coefficient will be calculated for each frame at every convolutional block, it will result in learned temporal weights in each video. To better illustrate this point, we conduct the experiment by removing Gumbel-Softmax [14] in our navigation module and modifying it to learn soft temporal weights for the features at focal branch. We can observe that AFNet(soft-weights) has a similar performance with AFNet(RT=0.25), AFNet(RT=0.50) and outperforms AFNet(RT=1.00) significantly which indicates that learning soft frame-wise weights causes the similar effect.

Table 2: Performance comparison on Something-Something (Sth-Sth) datasets. GFLOPs represents the average computation to process one video.

| Method | Dynamic | Backbone | Frames | Sth-Sth V1 | | Sth-Sth V2 | |
|---|---|---|---|---|---|---|---|
| | | | | Top-1 Acc. | GFLOPs | Top-1 Acc. | GFLOPs |
| TRN$_{RGB/Flow}$ [34] | ✗ | BN-Inception | 8/8 | 42.0% | 32.0 | 55.5% | 32.0 |
| ECO [35] | ✗ | BN-Inception+ResNet18 | 8 | 39.6% | 32.0 | - | - |
| TSM [20] | ✗ | ResNet50 | 8 | 45.6% | 32.7 | 59.1% | 32.7 |
| bLVNet-TAM [7] | ✗ | bLResNet50 | 16 | 47.8% | 35.1 | 60.2% | 35.1 |
| TANet [21] | ✗ | ResNet50 | 8 | 47.3% | 33.0 | 60.5% | 33.0 |
| SmallBig [18] | ✗ | ResNet50 | 8 | 47.0% | 52.0 | 59.7% | 52.0 |
| TEA [19] | ✗ | ResNet50 | 8 | 48.9% | 35.0 | 60.9% | 35.0 |
| SlowFast [8] | ✗ | ResNet50+ResNet50 | 8×8 | - | - | 61.7% | 66.6×3 |
| AdaFuse-TSM [24] | ✔ | ResNet50 | 8 | 46.8% | 31.5 | 59.8% | 31.3 |
| AdaFocus-TSM [28] | ✔ | MobileNetV2+ResNet50 | 8+12 | 48.1% | 33.7 | 60.7% | 33.7 |
| AFNet-TSM (RT=0.4) | ✔ | AF-ResNet50 | 12 | 49.0% | 27.9 | 61.3% | 27.8 |
| AFNet-TSM (RT=0.8) | ✔ | AF-ResNet50 | 12 | 49.9% | 31.7 | 62.5% | 31.7 |
| AFNet-TSM (RT=0.4) | ✔ | AF-MobileNetV3 | 12 | 45.3% | **2.2** | 58.4% | **2.2** |
| AFNet-TSM (RT=0.8) | ✔ | AF-MobileNetV3 | 12 | 45.9% | 2.3 | 58.6% | 2.3 |
| AFNet-TSM (RT=0.4) | ✔ | AF-ResNet101 | 12 | 49.8% | 42.1 | 62.5% | 41.9 |
| AFNet-TSM (RT=0.8) | ✔ | AF-ResNet101 | 12 | **50.1**% | 48.9 | **63.2**% | 48.5 |

**More is less.** We incorporate our method with temporal shift module (TSM [20]) to validate that AFNet can further reduce the redundancy of such competing methods and achieve *More is Less* by seeing more frames with less computation. We implement our method on Something-Something V1&V2 datasets which contain strong temporal information and relevant results are shown in Table 2.

Table 3: Comparisons with competitive efficient video recognition methods on Mini-Kinetics. AFNet achieves the best trade-off compared to existing works. GFLOPs represents the average computation to process one video.

| Method | Mini-Kinetics | |
|---|---|---|
| | Top-1 Acc. | GFLOPs |
| LiteEval [30] | 61.0% | 99.0 |
| SCSampler [16] | 70.8% | 42.0 |
| AR-Net [23] | 71.7% | 32.0 |
| AdaFuse [24] | 72.3% | 23.0 |
| AdaFocus [28] | 72.2% | 26.6 |
| VideoIQ [25] | 72.3% | 20.4 |
| AFNet (RT=0.4) | **72.8%** | **19.4** |
| AFNet (RT=0.8) | **73.5%** | 22.0 |

Table 4: Comparisons with competitive efficient video recognition methods on ActivityNet. AFNet achieves the best trade-off compared to existing works. GFLOPs represents the average computation to process one video.

| Method | ActivityNet | |
|---|---|---|
| | mAP | GFLOPs |
| AdaFrame [29] | 71.5% | 79.0 |
| LiteEval [30] | 72.7% | 95.1 |
| ListenToLook [9] | 72.3% | 81.4 |
| SCSampler [16] | 72.9% | 42.0 |
| AR-Net [23] | 73.8% | 33.5 |
| VideoIQ [25] | 74.8% | 28.1 |
| AdaFocus [28] | 75.0% | 26.6 |
| AFNet (RS=0.4,RT=0.8) | **75.6%** | **24.6** |

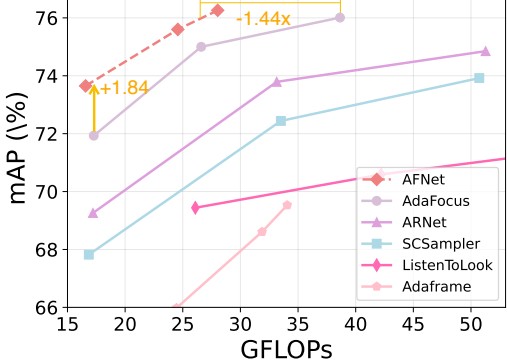

Figure 4: Computational cost vs mean Average Precision on ActivityNet. Our method is implemented with different ratio of selected frames $RT \in \{0.6, 0.8, 1.0\}$ and a fixed ratio of selected regions $RS \in \{0.4\}$.

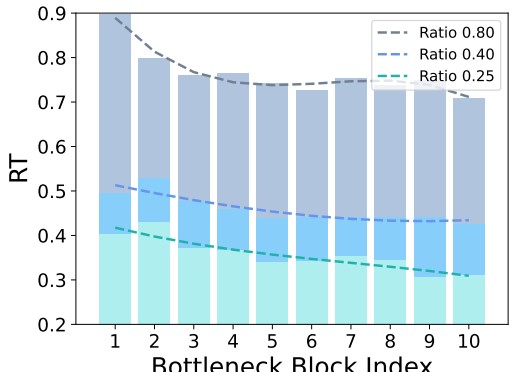

Figure 5: Distribution of $RT \in \{0.25, 0.4, 0.8\}$ for each convolution block in AF-ResNet50 on Something-Something V1. Dash lines denote using 3rd-order polynomials to estimate the trend of distribution.

Compared to TSM which samples 8 frames, our method shows significant advantages in performance as we introduce more frames and the two-branch structure can preserve the information of all frames. Yet, our computational cost is much smaller than TSM as we allocate frames with different computation resources by this two-branch design and adaptively skip the unimportant frames with the proposed navigation module. Moreover, AFNet outperforms many static methods, which carefully design their structures for better temporal modeling, both in accuracy and efficiency. This can be explained by that the navigation module restrains the noise of unimportant frames and enforces frame-wise attention which is beneficial for temporal modeling. As for other competitive dynamic methods like AdaFuse and AdaFocus, our method shows an obviously better performance both in accuracy and computations. When costing similar computation, AFNet outperforms AdaFuse and AdaFocus by 3.1% and 1.8% respectively on Something-Something V1. Furthermore, we implement our method on other backbones for even higher accuracy and efficiency. When we build AFNet on efficient structure MobileNetV3, we can obtain similar performance with TSM but only with the computation of 2.3 GFLOPs. Besides, AFNet-TSM(RT=0.8) with the backbone of ResNet101 can achieve the accuracy of 50.1% and 63.2% on Something-Something V1 and V2, respectively, which further validate the effectiveness and generalization ability of our framework.

**Comparisons with competitive dynamic methods.** Then, we implement our method on Mini-Kinetics and ActivityNet, and compare AFNet with other efficient video recognition approaches. At first, we validate our method on Mini-Kinetics and AFNet shows the best performance both in accuracy and computations compared with other efficient approaches in Table 3. To demonstrate that AFNet can further reduce spatial redundancy, we extend the navigation module to select salient

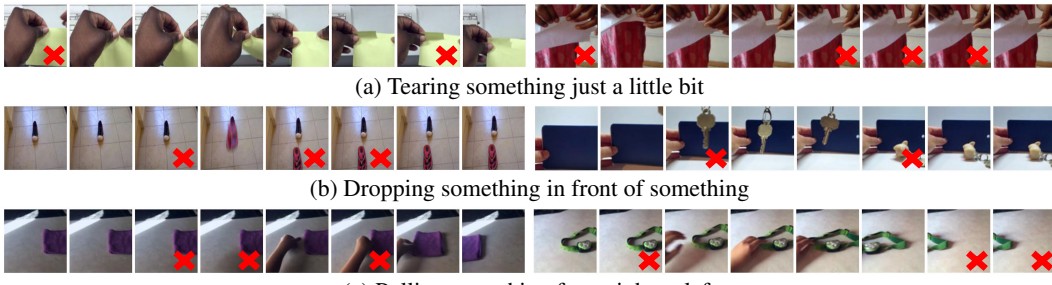

(a) Tearing something just a little bit

(b) Dropping something in front of something

(c) Pulling something from right to left

Figure 6: Visualization results on Something-Something V1 dataset. Frames annotated with red crosses denote the ones that AFNet does not select.

Table 5: Ablation study on navigation module and two-branch architecture. Different sampling policies are compared with our dynamic temporal selection module in various selection ratios.

| Ablation | | | mAP | | |
|---|---|---|---|---|---|
| Structure | Temporal Sampling Policy | Spatial Sampling Policy | 0.25 | 0.50 | 0.75 |
| Single Branch | Navigation Module | - | 46.8% | 60.4% | 71.3% |
| Two Branch | Random Sampling | - | 62.8% | 71.3% | 74.2% |
| | Uniform Sampling | | 59.0% | 71.7% | 74.7% |
| | Normal Sampling | | 61.5% | 71.4% | 74.6% |
| Two Branch | Navigation Module | - | **72.2%** | **74.3%** | **75.8%** |
| Two Branch | - | Random Sampling | 69.2% | 71.4% | 74.1% |
| | - | Center Cropping | 68.9% | 72.0% | 73.9% |
| Two Branch | - | Navigation Module | **72.0%** | **74.3%** | **75.9%** |

regions of important frame on ActivityNet. We move the temporal navigation module to the first layer of the network to avoid huge variance in features when incorporating spatial navigation module and note that we only apply this procedure in this part. We can see from Table 4 that our method obtains the best performance while costing the least computation compared to other works. Moreover, we change the ratio of selected frames and plot the mean Average Precision and computational cost of various methods in Figure 4. We can conclude that AFNet exhibits a better trade-off between accuracy and efficiency compared to other works.

## 4.2 Visualizations

We show the distribution of RT among different convolution blocks under different selection ratios in Figure 5 and utilize 3rd-order polynomials to display the trend of distribution (shown in dash lines). One can see a decreased trend in RT for all the curves with the increased index in convolution blocks and this can be explained that earlier layers mostly capture low-level information which has relatively large divergence among different frames. While high-level semantics between different frames are more similar, therefore AFNet tends to skip more at later convolution blocks. In Figure 6, we visualize the selected frames in the 3rd-block of our AFNet with $RT$=0.5 on the validation set of Something-Something V1 where we uniformly sample 8 frames. Our navigation module effectively guides the focal branch to concentrate on frames which are more task-relevant and deactivate the frames that contain similar information.

## 4.3 Ablation Study

In this part, we implement our method on ActivityNet with 12 sampled frames to conduct comprehensive ablation study to verify the effectiveness of our design.

**Effect of two branch design.** We first incorporate our navigation module into ResNet50 and compare it with AFNet to prove the strength of our designed two-branch architecture. From Table 5, AFNet shows substantial advantages in accuracy under different ratios of select frames. Aside from it, models which adopted our structure but with a fixed sampling policy also show significantly better performance compared with the network based on single branch which can further demonstrate the effectiveness of our two-branch structure and the necessity to preserve the information of all frames.

**Effect of navigation module.** In this part, we further compare our proposed navigation module with three alternative sampling strategies in different selection ratios: (1) random sampling; (2) uniform sampling: sample frames in equal step; (3) normal sampling: sample frames from a standard gaussian distribution. Shown in Table 5, our proposed strategy continuously outperforms other fixed sampling policies under different selection ratios which validates the effectiveness of the navigation module. Moreover, the advantage of our method is more obvious when the ratio of selected frames is small which demonstrates that our selected frames are more task-relevant and contain essential information for the recognition. Further, we evaluate the extension of navigation module which can reduce spatial redundancy, and compare it with: (1) random sampling; (2) center cropping. Our method shows better performance compared with fixed sampling strategies under various selection ratios which verifies the effectiveness of this design.

## 5 Conclusion

This paper proposes an adaptive Ample and Focal Network (AFNet) to reduce temporal redundancy in videos with the consideration of architecture design and the intrinsic redundancy in data. Our method enables 2D-CNNs to have access to more frames to look broadly but with less computation by staying focused on the salient information. AFNet exhibits promising performance as our two-branch design preserves the information of all the input frames instead of discarding part of the knowledge at the beginning of the network. Moreover, the dynamic temporal selection within the network not only restrains the noise of unimportant frames but enforces implicit temporal modeling as well. This enables AFNet to obtain even higher accuracy when using fewer frames compared with static method without temporal modeling module. We further show that our method can be extended to reduce spatial redundancy by only computing important regions of the selected frames. Comprehensive experiments have shown that our method outperforms competing efficient approaches both in accuracy and computational efficiency.

## Acknowledgments and Disclosure of Funding

Research was sponsored by the DEVCOM Analysis Center and was accomplished under Cooperative Agreement Number W911NF-22-2-0001. The views and conclusions contained in this document are those of the authors and should not be interpreted as representing the official policies, either expressed or implied, of the Army Research Office or the U.S. Government. The U.S. Government is authorized to reproduce and distribute reprints for Government purposes notwithstanding any copyright notation herein.

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
