## A. Experimental Settings

**ImageNet.** We first train our backbone network on ImageNet [2] using SGD optimizer. The L2 regularization coefficient and momentum are set to 0.0001 and 0.9 respectively. We train the network for 90 epochs with a batch size of 256 on 2 NVIDIA Tesla V100 GPUs and adopt a 5-epoch warmup procedure. The initial learning rate is set to 0.1 and it decays by 0.1 at epochs 30 & 60.

**Mini-Kinetics and ActivityNet.** Then we add navigation module and train it along with the backbone network on video datasets. On Mini-Kinetics [5] and ActivityNet [1], we use SGD optimizer with a momentum of 0.9 and the L2 regularization coefficient is set to 0.0001. The initial learning rate is set to 0.002 and it will decay by 0.1 at epochs 20 & 40. The models are trained for 50 epochs with a batch size of 32 on 2 NVIDIA Tesla V100 GPUs. The loss factor $\lambda$ is set to 1 for the two datasets and the temperature $\tau$ decreases from 1 to 0.01 exponentially.

**Jester and Something-Something.** The training details on Jester [8] and Something-Something [3] datasets are the same with ActivityNet [1] except the following changes: the initial learning rate is 0.01 which decays at 25 & 45 epochs and it is trained for 55 epochs in total; the loss factor $\lambda$ on these datasets is set to 0.5; the training data will be resized to $240 \times 320$ and then cropped to $224 \times 224$ as the raw data on these two datasets has relatively smaller resolutions.

## B. Building AFNet on BasicBlock

Table 1: Comparisons with baseline method on Jester.

| Method | Frames | Jester Top-1 Acc. |
|---|---|---|
| $\text{TSN}^{R34}$ [12] | 8 | 83.5% |
| $\text{AFNet}^{R34}$ (RT=0.75) | 8 | **89.3**% |
| $\text{AFNet}^{R34}$ (RT=0.50) | 8 | **89.7**% |
| $\text{AFNet}^{R34}$ (RT=0.25) | 8 | **89.5**% |

In previous experiments, we build AFNet on ResNet50 [4] which is made up of Bottleneck structure. Instead, we build AFNet with BasicBlock in this part on Jester dataset and compare it with baseline method TSN [12]. Table 1 shows that our method continuously shows significant advantages over TSN [12] in different selection ratios which validates the effectiveness of our method on BasicBlock structure as well. Interestingly, AFNet obtains the best performance when the selection ratio is set to 0.5 and it shows relatively the lowest accuracy when selecting more frames. This can be explained that our navigation module effectively restrains the noise of meaningless frames and implements implicit temporal modeling which utilizes fewer frames but obtains higher accuracy.

## C. Building AFNet wtih More Frames

Table 2: Comparisons with baseline method on ActivityNet with more sampled frames.

| Method | Frames | ActivityNet | |
|---|---|---|---|
| | | mAP. | GFLOPs |
| TSN [12] | 16 | **76.9**% | 62.8G |
| AR-Net [9] | 16 | 73.8% | 33.5G |
| VideoIQ [10] | 16 | 74.8% | 28.1G |
| AdaFocus [13] | 16 | 75.0% | **26.6G** |
| AFNet (RS=0.4,RT=0.8) | 16 | 76.6% | 32.9G |
| TSN [12] | 32 | **78.0**% | 131.2G |
| AFNet (RS=0.4,RT=0.8) | 32 | 77.6% | **60.9G** |

We build AFNet with more sampled frames in this section and compare it with baseline method. The results are shown in Table 2. When sample 16 frames, TSN exhibits a clear advantage in performance over other efficient methods which can be explained by the information loss in the preprocessing phase (e.g., frame selection, patch cropping) of these dynamic methods. This phenomenon motivates us to design AFNet, which adopts a two-branch structure to prevent the loss of information. The results show that AFNet costs significantly less computation compared to baseline method with only a slight drop in performance. Furthermore, we conduct experiments on 32 frames and the phenomenon is similar to 16 frames.

## D. More Ablation of AFNet

Table 3: Ablation of Fusion Strategy and Temperature Decay Schedule on ActivityNet.

| Fusion Strategy | Temperature Decay Schedule | mAP RT=0.5 |
|---|---|---|
| Addition | Exponential | 73.5% |
| Dynamic Fusion | Cosine | 74.1% |
| Dynamic Fusion | Linear | 73.8% |
| Dynamic Fusion | Exponential | **74.3**% |

We further include more ablation of AFNet on ActivityNet with 12 sampled frames. First, we test AFNet's performance without the dynamic fusion module and the results in Table 3 can demonstrate that this design is nontrivial as it effectively balances the weights between the features from two branches. Besides, we explore different temperature decay schedules, including: 1) decay exponentially, 2) decay with a cosine shape, 3) decay linearly. The results show that exponential decay achieves the best performance and we adopt this as the default setting in all our experiments.

## E. Limitations and Potential Negative Societal Impacts

First, the backbone of AFNet needs to be specially trained on ImageNet because of the two branch structure, while most other methods directly utilize the pretrained ResNet [4] from online resources. To make AFNet can be conveniently used by others, we offer the pretrained backbone on ImageNet which can be accessed in our provided code. Second, we did not consider build any temporal modeling module during the design of AFNet which is the main focus of other static methods, like TEA [6], TDN [11], etc. However, we have demonstrated that AFNet implements implicit temporal modeling and it is compatible with existing temporal modeling module, like TSM [7]. To the best of our knowledge, our method has no potential negative societal impacts.