# OpenReview forum: "Look More but Care Less in Video Recognition"
_NeurIPS.cc/2022/Conference — NeurIPS 2022 Accept_

### Official Review · Reviewer_bMpQ · 2022-07-08

**Rating:** 5
**Confidence:** 4
**Soundness:** 3 good
**Presentation:** 2 fair
**Contribution:** 2 fair

**Summary:**

The paper proposed to reduce the cost and boost the accuracy of CNN models applied to video action recognition via a two-stream approach. The first, "ample" stream processes all of the frames, but cheaply, by using low spatial resolution and number of channels. The second "focal" stream processes frames at high-resolution, but only processes a few of the input frames. A navigation model uses the input from the ample stream to select which frames the focal stream should process, using Gumbel softmax.

The paper presents results on the ActivityNet, Something-Something, and Mini-Kinetics datasets, demonstrating strong accuracies at competitively-low flop counts.

Additional studies explore allowing the navigation module to make spatial, as well temporal, selections, and demonstrate that the learned navigation model achieves superior performance to other simpler sampling strategies.


**Questions:**

Why is TSN used as a baseline in Table 1? Although it's a great paper, the method is 6 years old.


**Limitations:**

yes

**Strengths And Weaknesses:**

**Originality** The main weakness is that the premise has already been explored thoroughly in other publications. Specifically in "SlowFast Networks for Video Recognition" https://arxiv.org/abs/1812.03982 (ICCV 2019, 1300+ citations). Both use the idea of a high frame rate low spatial resolution / channels stream, combined with a low frame rate high spatial resolution, with lateral connections from the fast pathway to the slow pathway. Both are targeted at limiting flops while boosting accuracy.

More recently in "Multiview Transformers for Video Recognition" (CVPR 2022) which extends the slow-fast premise to 2+ streams using transformer backbones, instead of CNNs.

The paper and accompanying analysis is incomplete without a direct comparison to SlowFast.

The primary difference is the adaptive selection of frames by the navigation module, vs. SlowFast's fixed temporal sampling rates plus lateral connections. Given that, at a quick glance the numbers seem comparable between the two(*), additional comparison between the approaches is warranted.

**Clarity:** The paper is clearly written and generally easy to follow. Section 3.2 is a minor exception: I was eager to see the implicit temporal modeling, but found this explanation hard to follow. Perhaps it would benefit from more prose and fewer equations?

Line 293: Please elaborate on how to "sample frames from a gaussian distribution", which is not obvious since Gaussian variables are continuous, not discrete.

**Quality:** Generally good, except for the omission of highly-cited related work.

**Significance:** As mentioned, this paper's significance is diminished by its limited originality.


===

(*) The original SlowFast paper publishes numbers on different datasets than this work, but from what I can piece together, SlowFast seems better or comparable. If it's fair to compare numbers on MiniKinetics and Kinetics-400, SlowFast gets 74.2% top-1 at 28.6 GFLOPs (see SlowFast Table 2 (b)), while this paper reports 73.5% top-1 at 22.0 GFLOPs. On Something-Something v2, SlowFast-ResNet50 seems to get 61.7 (see "Multiview Transformers..." Table 2 (c)) which is comparable to this paper's 61.3/62.5.

I recognize the limitations of this analysis, and would love to see a proper apples-to-apples comparison.

---

> ### Author Response · Authors · 2022-08-02
> **Response to Reviewer bMpQ_Part 1**
>
> We appreciate the Reviewer's careful comments to point out our unclear description and we make the response as below.
>
>
> **Originality: Discussion with SlowFast[1].**
> AFNet looks similar to SlowFast as both adopt a two-branch design. Nevertheless, we want to stress that the motivation for the two-branch structure of the two works is totally different. For SlowFast, it is designed to capture semantic information and changing motion with branches at different temporal speeds for better performance. To achieve this, it adopts fixed temporal sampling rates on both branches and uses 3D convolutions for temporal modeling. As for AFNet, the focus of this paper is on efficiency, and the two-branch architecture is motivated by the phenomenon that most existing dynamic networks for video recognition will lead to information loss (more detailed analysis can be found in our reply to Weakness 1 pointed out by Reviewer ChDo). Therefore, we design a very lightweight ample branch to prevent the information lost in focal branch. Besides, we do not design any temporal modeling module like other static 2D methods TSM[2], TEA[3] which is the common practice in this area (see papers like AR-Net[4], AdaFocus[5], VideoIQ[6]). More discussion with SlowFast can be found in our reply to Weakness 1 pointed out by Reviewer VKUT. We thank the Reviewer for pointing this out and we will add the discussion in our revision.
>
> **Clarity: Implicit temporal modeling.**
> Sorry for the inconvenience brought in reading. We will try to clarify this part here. First, we want to reemphasize that the focus of our work is on efficiency, and we do not explicitly design any temporal modeling module like other 2D methods TSM, TEA. Therefore, a fair baseline for AFNet should be TSN[7] as both methods simply average the predictions of each frame to present the final prediction of the corresponding video.
>
> Though, we demonstrate that AFNet enforces implicit temporal modeling by the navigation module. In Equation (14), $L_n$ is a binary temporal mask which will decide whether the coefficient will be calculated for each frame at every convolutional block.
> For each stage, it is made up of multiple convolutional blocks and each frame at every block will be assigned a binary weight (1 or 0) by the temporal mask $L_n$. Though there is only a binary choice for the weights at each block, the weights will approximate soft temporal weights if we consider the whole stage in a long run. Moreover, in Equation (3), the logits $p_n^t$ for mask generation are produced by $W_2$ which models the relations between frames.
> Therefore, the temporal mask $L_n$ has taken the temporal information into account and the series multiplication of the output of each convolutional block results in learnable temporal weights, which we regard as implicit temporal modeling.
> This part can be proved by the experiments in Table 1 as AFNet exhibits much higher accuracy compared with TSN when selecting fewer frames.
> However, if all the frames are chosen, all temporal masks will be filled with the value of 1 which does not cause a temporal divergence between frames.
>
> **Clarity: Sampling from Gaussian distribution.**
> Thanks for noticing the details in our experiments. Assume we will sample $N$ frames for each video. First, we build a gaussian function with the mean value of $(N+1)/2$ and the standard deviation value of $N/6$. Then, we will get the probabilities of sampling each frame by inputting the frame index from $1$ to $N$ to the gaussian function. After that, we have two implementations for the sampling procedure. The first one is to select top-k frames based on the probabilities we get. However, the selected frames are fixed as the probabilities do not change. Therefore, we have introduced Gumbel softmax[8] to approximate the probabilities from the gaussian function and then select the top-k frames based on the generated probabilities. Note that the generated probabilities are not fixed as it contains Gumbel noise. We have tested both methods and the second implementation achieves slightly better performance, so we reported their results in Table 5.

---

> ### Author Response · Authors · 2022-08-02
> **Response to Reviewer bMpQ_Part 2**
>
> **Significance: Comparison with SlowFast.**
> First, we want to stress that the main focus of our paper is on efficiency and the main comparisons in our paper should be with other efficient dynamic networks which is the common practice of this area [4],[5],[6]. Therefore, we conduct experiments on datasets like Mini-Kinetics, ActivityNet instead of Kinetics-400, Kinetics-600 in order to compare with other dynamic methods.
>
> The result of SlowFast-ResNet50 on Something-Something v2 in Multiview Transformer[9] comes from Table 5 in Multigrid training[10]. The accuracy of 61.7 is not the baseline result but the result of the model trained with Multigrid training. Therefore, we should use the accuracy of 60.9 for a fair comparison. Besides, it is included in [10] that the model uses the speed ratio a=4 and channel ratio of b=1/8 with 64 frames on the fast pathway. Therefore, there will be 16 frames on slow pathway and the GFLOPs should be much greater than 36.1 (channel ratio of b=1/8, 4 frames on slow pathway, and 32 frames on fast pathway). Moreover, we have discovered from the official website of PyTorch video (https://pytorchvideo.readthedocs.io/en/latest/model_zoo.html) that SlowFast-R50 achieve the accuracy of 61.68 with a GFLOPs of 66.60x3. We conclude the results on Something-Something v2 in this table:
> | Method  | Pretrain | Top-1 Acc. |  GFLOPs |
> | :-----: | ------------ | :-----: | :-----: |
> | SlowFast | Kinetics400 | 60.9% | >36.1G |
> | SlowFast | Kinetics400 | 61.7% | 66.6x3G |
> | AFNet-TSM(RT=0.4) | ImageNet | 61.3% | 27.8G |
> | AFNet-TSM(RT=0.8) | ImageNet | 62.5% | 31.7G |
>
> We can observe that AFNet-TSM exhibits clearly better performance on this dataset in both accuracy and efficiency. Furthermore, SlowFast is pretrained on Kinetics400, and AFNet is pretrained on ImageNet. We believe our method will obtain higher accuracy if we pretrain on Kinetics400. We thank the Reviewer for pointing it out and we will include this part in our final version.
>
> **Question: Why compare with TSN?**
> Thanks for this question. As we have explained in reply to implicit temporal modeling, the reason we compare with TSN is that AFNet does not design any temporal modeling module which is the same as TSN and both methods use ResNet-50[11] as the backbone for classification. Table 1 is used to prove that the navigation module enforces implicit temporal modeling in AFNet as it exhibits much higher accuracy compared with TSN when selecting fewer frames. With this design, we can achieve less is more: seeing fewer frames with higher accuracy.
>
>
> We hope the explanations can address your concerns and we would appreciate it a lot if you can recognize the contributions of our work.
>
>
> [1] Feichtenhofer C, Fan H, Malik J, et al. Slowfast networks for video recognition[C]//Proceedings of the IEEE/CVF international conference on computer vision. 2019: 6202-6211.
> [2] Lin J, Gan C, Han S. Tsm: Temporal shift module for efficient video understanding[C]//Proceedings of the IEEE/CVF International Conference on Computer Vision. 2019: 7083-7093.
> [3] Li Y, Ji B, Shi X, et al. Tea: Temporal excitation and aggregation for action recognition[C]//Proceedings of the IEEE/CVF conference on computer vision and pattern recognition. 2020: 909-918.
> [4] Meng Y, Lin C C, Panda R, et al. Ar-net: Adaptive frame resolution for efficient action recognition[C]//European Conference on Computer Vision. Springer, Cham, 2020: 86-104.
> [5] Wang Y, Chen Z, Jiang H, et al. Adaptive focus for efficient video recognition[C]//Proceedings of the IEEE/CVF International Conference on Computer Vision. 2021: 16249-16258.
> [6] Sun X, Panda R, Chen C F R, et al. Dynamic network quantization for efficient video inference[C]//Proceedings of the IEEE/CVF International Conference on Computer Vision. 2021: 7375-7385.
> [7] Wang L, Xiong Y, Wang Z, et al. Temporal segment networks: Towards good practices for deep action recognition[C]//European conference on computer vision. Springer, Cham, 2016: 20-36.
> [8] Jang E, Gu S, Poole B. Categorical reparameterization with gumbel-softmax[J]. arXiv preprint arXiv:1611.01144, 2016.
> [9] Yan S, Xiong X, Arnab A, et al. Multiview transformers for video recognition[C]//Proceedings of the IEEE/CVF Conference on Computer Vision and Pattern Recognition. 2022: 3333-3343.
> [10] Wu C Y, Girshick R, He K, et al. A multigrid method for efficiently training video models[C]//Proceedings of the IEEE/CVF Conference on Computer Vision and Pattern Recognition. 2020: 153-162.
> [11] He K, Zhang X, Ren S, et al. Deep residual learning for image recognition[C]//Proceedings of the IEEE conference on computer vision and pattern recognition. 2016: 770-778.

---

> ### Author Response · Authors · 2022-08-06
> **Sincerely looking forward to your further feedback!**
>
> Dear Reviewer bMpQ,
>
> Thank you so much for helping improve our paper so far!
>
> Given the NeurIPS final discussion deadline (08/09) is approaching, we really hope to have a further discussion with you to see if our responses solve your concerns.
>
> Thank you so much for being with us so far! Have a wonderful weekend!
>
> Sincerely,
> Authors

---

> ### Author Response · Authors · 2022-08-09
> **Gentle reminder for the deadline**
>
> Dear Reviewer bMpQ:
>
> Thank you so much for delivering your valuable comments to us. We have tried our best to provide a detailed response to you and revised the draft accordingly based on your concerns.
>
> We would like to kindly remind you that tomorrow (Aug 9th) is the deadline for the discussion phase and your opinions regarding our response matter a lot to us.
>
> Thank you so much for helping us improve our paper.
>
> Sincerely,
> Authors

---

> > ### Author Response · Authors · 2022-08-09
> > **updated draft**
> >
> > Dear Reviewer bMpQ:
> >
> > We have submitted the final version of our draft just now. Compared with the previous version, we have rewritten the explanations for implicit temporal modeling in Section 3.2 and included the reasons for comparisons with TSN in Table 1.
> >
> > In the previous version, we have already added the discussion with SlowFast to emphasize the difference and compared AFNet with SlowFast on Something-Something V2 in Table 2 where our method exhibits advantages both in accuracy and efficiency.
> >
> > Thank you so much for providing valuable comments to us and helping us improve our paper!
> >
> > Sincerely,
> > Authors

---

### Official Review · Reviewer_ztLC · 2022-07-10

**Rating:** 4
**Confidence:** 5
**Soundness:** 2 fair
**Presentation:** 2 fair
**Contribution:** 2 fair

**Summary:**

This paper proposes Ample and Focal Network (AFNet) for video recognition. Specifically, the network has an ample and a focal branch. The ample branch operates on a set of neighboring (with strides) frames with reduce-sized feature maps (in height, width and channels), whereas the focal networks then takes in both input frames and intermediate predictions from focal network to deceptively process selected frames, with higher computation budget. The resulting network is claimed to have a better accuracy-computation trade-off than previous work. The authors conducted experiments on Something-Something v1/v2, Mini-Kinetics, Jester and ActivityNet and demonstrated that their method yields better accuracy compared to several baselines (e.g. TSM, AdaFuse-TSM, bLVNet etc) on these datasets.


**Questions:**

- L150, isn't L_n a continuous vector computed using Eq 6? Is there some thresholding used here before selecting non-zero values?
- Eq 1, is it necessary to keep the v notation on the left hand side of the equation? Since it represents input video, keeping it here might cause some confusion.
- L145, "we let tau decrease from 1 to 0.01 during train", what's the decay schedule used? Do different schedulings make a difference?
- Table 1, the improvement from TSN to AFNet is quite significant, and honestly a bit surprising. Does the author investigate the possibility of overfitting for the TSN results?


**Limitations:**

Yes

**Strengths And Weaknesses:**

Strengths
+ The two-branch design makes intuitive sense, with one focus on lightweight processing on dense inputs, while the other process sparsely selected inputs with heavier computation.
+ The paper is comprehensive in structure --- in addition to the qualitative description of the approach, the authors also included a section on the theoretical analysis of implicit temporal modeling.
+ The result section includes comparison to several baseline methods across several datasets. It seems the proposed method has an edge on accuracy-computation trade-off across these comparisons. There are also some qualitative visualizations and ablations to dissect the approach.

Weaknesses
- As shown in Figure 2, the frame selection process is implemented as sparse convolutions, for which from the texts I cannot tell how efficient they are. This becomes more of an issue since in all tables the authors report FLOPs rather than actual inference latency.
- Across Sth-Sth (Table 2), Mini-Kinetics (Table 3), and ActivityNet (Table 4), the accuracy/computation gains over the competitor is definitely not to a level that I'll consider significant.
- There is no promise on code release, which might make it hard to reproduce the reported results.
- For related work, the proposed approach missed an important citation: SlowFast Networks from Feichtenhofer et al. As it also sits on this two-branch idea for video recognition where one lightweight branch focuses on motion and another heavy branch focuses on semantics. This should be added and discussed. Another related paper is "End-to-end Learning of Action Detection from Frame Glimpses in Videos" from Yeung et al., as it first proposed to selective focus on a subset of frames for video recognition.

Writing/Typos
- L58, "but strength the representation" --> "but strengthen the representation"
- L125, C_o, H_o and W_o are not introduced anywhere up until this point.
- L273, "to analysis the results" --> "to analyze the results"
- In L136, t in p^t_n is used to index frames, whereas in Eq 4, the superscript is overloaded to denote the frame selection flag. This will cause some confusions.

---

> ### Author Response · Authors · 2022-08-02
> **Response to Reviewer ztLC_Part 1**
>
> We thank for the Reviewer pointing out the drawback of our draft and we make the response as below.
>
> **Weakness 1: Practical Speedup.**
> We thank the Reviewer for this comment. First, we want to clarify that the frame selection process in our method is not technically implemented as sparse convolution. As AFNet is 2D based model which uses 2D convolutions, the temporal dimension is put on batch dimension in real implementation. Therefore, the frame selection can be easily implemented by the slicing operation on batch dimension and this process will not hurt the computational graph of vanilla convolution, unlike other pruning methods. In this way, we can achieve efficiency by only computing the selected frames at each convolutional block.
>
> Further, we test the CPU (Intel(R) Core(TM) i7-6850K CPU @ 3.60GHz) and GPU (NVIDIA GeForce GTX TITAN X) inference time of the competing methods listed in Table 2. Note that all methods are implemented on ResNet-50[1] and we use the original code provided by the authors. We sample 12 frames with the input size of 224x224 for all methods for fair comparison and get the average inference time over 100 runs:
> | Method  | GPU(ms) | CPU(ms) |
> | :-----: | ------------ | :-----: |
> | bLVNet-TAM | 37 | 798 |
> | TANet | 27 | 595 |
> | SmallBig | 66 | 1268 |
> | TEA | 40 | 755 |
> | TSM | 22 | 633 |
> | AdaFocus-TSM | 45 | 744 |
> | AdaFuse-TSM | N/A | N/A |
> | AFNet-TSM(training) | 52 | 518 |
> | AFNet-TSM(inference) | 32 | 422 |
>
>
> We can see from the table that AFNet achieves the fastest speed on CPU, making it favorable for employment on edge devices. While the GPU speed of AFNet is not as good as the speedup on CPU which shows inferior performance to static method TSM and TANet. The potential explanation is that the two-branch structure is less favorable in GPU acceleration and we do not have hardware-oriented optimization in our implementation yet. However, AFNet-TSM costs less inference time than dynamic method AdaFocus-TSM in both CPU and GPU. As for another dynamic method AdaFuse, the author does not provide the code for efficient inference, so we do not test their speed. Further, we compare the speed of AFNet-TSM in training mode (simply adding a temporal mask) with inference mode (only computing on salient frames) to demonstrate that the frame selection indeed boosts efficiency (more detailed explanations of the two modes can be found in our reply to Question 3 pointed out by Reviewer ChDo).
>
> **W2: Accuracy/computation gain not significant.**
> First, we want to stress that we did not make the claim that our method is significantly better than other approaches as previous works already push the line to a very high standard. However, we want to emphasize that our work offers a new perspective from other dynamic methods as the two-branch structure prevents the information loss (detailed analysis can be found in our reply to Weakness 1 pointed out by Reviewer ChDo). Moreover, the frame selection at intermediate features not only enforces implicit temporal modeling which is seldom touched in this area, but enables end-to-end training as well. This is also an advantage over many other dynamic methods like AR-Net[2], VideoIQ[3] as they have to split the training into multiple stages which makes them hard to implement.
>
> **W3: No promise on code release.**
> Thanks for bringing it up. We have provided the code during submission and will release the code once this paper is accepted.
>
> **W4: Missing related works.**
> We thank the Reviewer for pointing this out.
> However, the differences between SlowFast and AFNet are: (1) network category: SlowFast[4] is a static 3D model, but AFNet is a dynamic 2D network;
> (2) motivation: SlowFast is designed to capture semantic information and changing motion with branches at different temporal speed for better performance, while AFNet is aimed to dynamically skip frames to save computation and the design of two-branch structure is to prevent the information loss;
> (3) specific design: AFNet is designed to downsample features for efficiency at ample branch while SlowFast processes features in the original resolution;
> (4) temporal modeling: SlowFast applies 3D convolutions for temporal modeling, AFNet is a 2D model which is enforced with implicit temporal modeling by the designed navigation module.
> We will add this part in our final version and also cite "End-to-end Learning of Action Detection from Frame Glimpses in Videos"[5].
>
> **W5: Writing/Typos.**
> We will revise the typos and modify the mentioned sentence for clear descriptions. Thanks for the advice.

---

> ### Author Response · Authors · 2022-08-02
> **Response to Reviewer ztLC_Part 2**
>
>
> **Question 1: Generation of L_n.**
> $L_n$ is a binary mask drawn from the distribution $\pi$ shown in Equation (5). Specifically, we generate logits $p_n^t$ ($1 \times 2\times T \times 1 \times 1$) for each video to decide whether to choose each frame and use argmax to make it a one-hot tensor. Then, we slice the first dimension of the one-hot tensor to get the temporal mask $L_n$ ($1 \times T \times 1 \times 1$). As argmax is non-differentiable, we adopt Gumbel softmax[6] to allow discrete decisions in forward propagation and estimate gradients in backward propagation.
>
> **Q2: Revision on Equation (2).**
> Thanks for the suggestion and we will revise it in our final version.
>
>
> **Q3: Temperature decay schedule.**
> In all experiments, we let the temperature decay exponentially and we compare it with two variants including decay with a cosine shape and decay linearly on the dataset of ActivityNet:
> | Method  | mAP |
> | :-----: | ------------ |
> | AFNet(RT=0.5)_exp | 74.3% |
> | AFNet(RT=0.5)_linear | 73.8% |
> | AFNet(RT=0.5)_cos | 74.1% |
>
> The results show that our strategy obtain the highest accuracy compared to the other two variants and we will add this part in our final version.
>
> **Q4: Results in Table 1 are surprising.**
> The reason for the significant improvement is that AFNet enforces implicit temporal modeling compared with TSN[7] which does not build any temporal modeling module.
> In Equation (14), $L_n$ is a binary temporal mask which will decide whether the coefficient will be calculated for each frame at every convolutional block.
> For each stage, it is made up of multiple convolutional blocks and each frame at every block will be assigned a binary weight (1 or 0) by the temporal mask $L_n$. Though there is only a binary choice for the weights at each block, the weights will approximate soft temporal weights if we consider the whole stage in a long run. Moreover, in Equation (3), the logits $p_n^t$ for mask generation are produced by $W_2$ which models the relations between frames.
> Therefore, the temporal mask $L_n$ has taken the temporal information into account and the series multiplication of the output of each convolutional block results in learnable temporal weights, which we regard as implicit temporal modeling.
> Therefore, the results in Table 1 can be regarded as proof of our analysis in Section 3.2. Moreover, we have tested to modify the navigation module and change it to learn a soft temporal weight for each frame which obtain similar results in Table 1. This part can be found in our reply to Question 2 pointed out by Reviewer VKUT.
>
>
> We hope the explanations can address your concerns and we would appreciate it a lot if you can recognize the contributions of our work.
>
>
> [1] He K, Zhang X, Ren S, et al. Deep residual learning for image recognition[C]//Proceedings of the IEEE conference on computer vision and pattern recognition. 2016: 770-778.
> [2] Meng Y, Lin C C, Panda R, et al. Ar-net: Adaptive frame resolution for efficient action recognition[C]//European Conference on Computer Vision. Springer, Cham, 2020: 86-104.
> [3] Sun X, Panda R, Chen C F R, et al. Dynamic network quantization for efficient video inference[C]//Proceedings of the IEEE/CVF International Conference on Computer Vision. 2021: 7375-7385.
> [4] Feichtenhofer C, Fan H, Malik J, et al. Slowfast networks for video recognition[C]//Proceedings of the IEEE/CVF international conference on computer vision. 2019: 6202-6211.
> [5] Yeung S, Russakovsky O, Mori G, et al. End-to-end learning of action detection from frame glimpses in videos[C]//Proceedings of the IEEE conference on computer vision and pattern recognition. 2016: 2678-2687.
> [6] Jang E, Gu S, Poole B. Categorical reparameterization with gumbel-softmax[J]. arXiv preprint arXiv:1611.01144, 2016.
> [7] Wang L, Xiong Y, Wang Z, et al. Temporal segment networks: Towards good practices for deep action recognition[C]//European conference on computer vision. Springer, Cham, 2016: 20-36.

---

> ### Author Response · Authors · 2022-08-06
> **Sincerely looking forward to your further feedback!**
>
> Dear Reviewer ztLC,
>
> Thank you so much for helping improve our paper so far!
>
> Given the NeurIPS final discussion deadline (08/09) is approaching, we really hope to have a further discussion with you to see if our responses solve your concerns.
>
> Thank you so much for being with us so far! Have a wonderful weekend!
>
> Sincerely,
> Authors

---

> ### Author Response · Authors · 2022-08-09
> **Gentle reminder for the deadline**
>
> Dear Reviewer ztLC:
>
> Thank you so much for delivering your valuable comments to us. We have tried our best to provide a detailed response to you and revised the draft accordingly based on your concerns.
>
> We would like to kindly remind you that tomorrow (Aug 9th) is the deadline for the discussion phase and your opinions regarding our response matter a lot to us.
>
> Thank you so much for helping us improve our paper.
>
> Sincerely,
> Authors

---

### Official Review · Reviewer_aYAw · 2022-07-12

**Rating:** 5
**Confidence:** 4
**Soundness:** 3 good
**Presentation:** 3 good
**Contribution:** 3 good

**Summary:**

This paper presents an efficient framework called Ample and Focal Network (AFNet) for video recognition. This framework consists of two branches: the ample branch preserves all input features by lightweight computation; the focal branch extracts features only from selected frames to save cost. By fusing the features of these two branches, AFNet can keep its focus on the crucial information while requiring less computation. Experiments on five datasets demonstrate the superiority of AFNet compared to state-of-the-art methods. However, I have some concerns about this paper. My detailed comments are as follows.

**Questions:**

1. The proposed method uses several navigation modules in a recurrent manner. What did the model learn to select in different stages?
2. In section 3.1, the logit $p_n^t$ for frame t is generated with Eq.(3). Then for $p_n={p_n^t}_{t=1}^T$, all $p_n^t$ are with the same values as they are generated with the same convolution weights $W_2$ and feature $\tilde{v}_{y_n^a}$. More discussions are required.
3. The second term in Eq.(16) is used to contain the ratio of the selected frame with the square of the difference between $r$ and $RT$. However, This introduces an additional hyper-parameter $RT$ and restricts the model from selecting fewer frames for lower computation cost.
4. InTab.1, AFNet achieves higher accuracy with fewer frames, which is counterintuitive. The explanation given by the authors is only for why AFNet achieves higher accuracy than TSN, not for why the fewer frames are selected, the higher accuracy AFNet achieves. More clear explanations are required.
5. Why is AFNet not compared to other efficient frameworks like MoViNet[3]? More explanations are needed.
6. Some action recognition methods are missing, such as Two-Stream Network [1] and T-C3D [2].

[1]	“Convolutional Two-Stream Network Fusion for Video Action Recognition.” CVPR (2016)
[2]	“T-C3D: Temporal Convolutional 3D Network for Real-Time Action Recognition.” AAAI (2018).
[3]	"Movinets: Mobile video networks for efficient video recognition." CVPR (2021)



**Ethics Review Area:**

["I don’t know"]

**Limitations:**

The authors adequately addressed the limitations and potential negative societal impact of their work.

**Strengths And Weaknesses:**

Strengths:
1. This paper presents an efficient framework called Ample and Focal Network for video recognition, which uses more frames but reduces computation.
2. The authors use a two-branch framework, in which two branches are complementary to each other, to prevent information loss when selecting fewer frames.
3. The authors propose a navigation module that can select informative frames to save computational cost and is compatible with spatial adaptive works.

Weaknesses:
1. My biggest concern lies in the technical contribution of this paper. This method uses two streams, one for dealing with frames with high spatial resolution but low temporal resolution. The other process frames with low spatial resolution and high temporal resolution. Such an idea seems similar compared with the Slow-Fast [a] network.
2. As for the adaptive frame selection, Scsampler [b] also uses a tiny network to select frames and a larger network to do action recognition. I suggest adding more discussions between them.
3. In the experiments, the authors use only 12 frames (compared with 8-frame settings in previous methods), which is not convincing enough to verify the efficiency of the proposed methos.

[a] SlowFast Networks for Video Recognition. ICCV 2019.
[b] Compressing Videos to One Clip With Single-Step Sampling. CVPR 2022.

---

> ### Author Response · Authors · 2022-08-02
> **Response to Reviewer aYAw_Part 1**
>
> We appreciate the Reviewer's approval and valuable comments. We respond to the Reviewer's concerns as below.
>
> **Weakness 1: Technical Contribution.**
> Though AFNet seems similar to SlowFast[1], we stress that the motivation of the two-branch structure and specific designs of our method are different from it. AFNet is a dynamic 2D network which adaptively selects salient frames to achieve efficiency, while SlowFast is a static 3D model which captures semantic information and changing motion to obtain higher accuracy.
> Other differences are: (1) specific design: AFNet is designed to downsample features for efficiency at ample branch while SlowFast processes features in the original resolution;
> (2) temporal modeling: SlowFast applies 3D convolutions for temporal modeling, AFNet is a 2D model which is enforced with implicit temporal modeling by the designed navigation module. We thank the Reviewer for pointing it our and we will supplement this part in our final version.
>
> **W2: Discussion of SCSampler[12].**
> Indeed, SCSampler utilizes a tiny network to do frame selection and sends the selected salient frames into a larger network to do action recognition. Though, this process will lead to information loss as the unselected frames will be completely abandoned (more analysis can be found in our reply to Weakness 1 pointed out by Reviewer ChDo). However, we want to emphasize that our approach is different from this line of research as the frame selection process of AFNet is at intermediate features. This brings three effects:
> (1) the dynamic frame selection at intermediate features will empower the model with strong flexibility as different frames will be selected at different layers;
> (2) the temporal masks at different convolutional blocks will result in implicit temporal modeling which is demonstrated in Section 3.2 and the results in Table 1;
> (3) the information loss caused by frame selection on focal branch can be compensated by the features in the ample branch to prevent the information loss.
> We will include the discussion in our final version.
>
> **W3: Experiments with more frames.**
> We conduct experiments on ActivityNet with more sampled frames:
> | Method  | Frame | mAP |  GFLOPs |
> | :-----: | ------------ | :-----: | :-----: |
> | TSN | 16 | 76.9% | 65.6G |
> | AFNet(RS=0.4,RT=0.8) | 16 | 76.6% | 32.9G |
> | TSN | 32 | 78.0% | 131.2G |
> | AFNet(RS=0.4,RT=0.8) | 32 | 77.6% | 60.9G |
>
> When sampling more frames, AFNet costs significantly less computations compared to baseline method with only a slight drop in performance which can be attributed to two-branch structure as it avoids the information loss. Thanks for the advice and we will add this part in our revision.
>
> **Question 1: Learned knowledge at different stages.**
> Thanks for this question and we think it is important to analyze the learned policy to gain insight for the research community. In Figure 5, we calculate and show the distribution of RT at different stages. We can observe a decreased trend in RT for all curves which means that AFNet tends to select more frames at earlier layers and skip more at later stages. It can be explained that earlier layers mostly capture low-level information which has relatively large divergence among different frames. While high-level semantics between different frames are more similar, therefore AFNet tends to skip more at later convolution blocks.
>
> **Q2: Discussion on Equation (3).**
> The generated logits $p_n^t$ for each video have the shape as: $1\times(2\times T)\times1\times1$ which means that there will be two values assigned for each frame: $1\times2\times1\times1$. This vector denotes the possibility of whether to choose this frame and we send it to Gumbel Softmax module to generate a one-hot vector (which is the temporal mask $L$) for latter frame selection.
>
> **Q3: Hyperparameter RT.**
> The introduced hyperparameter RT is necessary for our method and we think it is the key for AFNet to select fewer frames for lower costs. By adding the second term in Equation (16), the network will be enforced to reduce $r$ (the ratio of selected frames in the network) to RT which is set before the training. For example, if we want to AFNet to achieve smaller costs, we can set RT to 0.3 so that $r$ will be forced to decrease from 1 to 0.3 in the training process. If we do not introduce RT and only use cross entropy loss, $r$ will remain nearly 1 during training as selecting all the frames potentially leads to better accuracy and smaller cross entropy loss which prevents AFNet to achieve smaller costs.

---

> ### Author Response · Authors · 2022-08-02
> **Response to Reviewer aYAw_Part 2**
>
> **Q4: Fewer frames, higher accuracy.**
> Thanks for pointing out this phenomenon. However, we want to clarify that we did not make the claim that selecting fewer frames will lead to higher accuracy. As we have mentioned in Section 3.2, the learned binary mask will decide a coefficient whether to be calculated for every frame at each block which results in learned temporal weights for implicit temporal modeling. Therefore, selecting a ratio of 0.25 and 0.5 frames will achieve much higher accuracy than selecting all frames (choosing all frames leads to a fixed weight for all frames). A possible explanation for the accuracy of AFNet(RT=0.25) being higher than AFNet(RT=0.5) is that a temporal mask which selects fewer frames can better restrain the noise of redundant frames.
>
> **Q5: Comparison with MoViNet.**
> Thanks for pointing it out. Though, we want to emphasize that this paper is aimed at developing efficient algorithms rather than building strong temporal modeling structures, and the main comparisons in this work should be AFNet with other dynamic methods, which is the common practice in this area (see papers like AR-Net[2], AdaFocus[3], VideoIQ[4]).
> Indeed, MoViNet[5] is a very efficient network for video recognition. However, it adopts 3D convolutions for temporal modeling, while AFNet is built on 2D network ResNet-50[6] which does not involve any temporal modeling module like other dynamic methods[2],[3],[4]. Besides, MoViNet is a NAS product which costs unaffordable computational cost in training for many researchers, while AFNet is a hand-crafted structure that can easily be trained in an end-to-end fashion like other static methods TSN[7], TSM[8]. Moreover, the search space of MoViNet is built on MobileNet v3[9], but AFNet is based on ResNet-50 which makes the comparison unfair.
> However, we are running experiments which build our designed AF module on MobileNet v3 to test the generalization of our method on efficient backbones. We will add the results in our revision.
>
> **Q6: Comparison with Two-stream network[10] and T-C3D[11].**
> As we mentioned in the reply to Question 5, the main comparisons in our paper should be with other dynamic methods and we only conduct experiments on datasets that are used in previous works[2],[3],[4]. In Two-stream Network and T-C3D, the results are only based on UCF101 and HMDB51. Therefore, we further conducted experiments on these two datasets and we report the mean accuracy across 3 splits in this table:
> | Method  | Pretrain | UCF101 Top-1 Acc. |  HMDB Top-1 Acc. |
> | :-----: | ------------ | :-----: | :-----: |
> | Two-stream(S:VGG-16 T:VGG-M) | Kinetics400 | 90.8% | 62.1% |
> | Two-stream(S:VGG-16 T:VGG-16)   | Kinetics400 | 92.5% | 65.4% |
> | T-C3D | Kinetics400 | 92.5% | 62.4% |
> | TSM | Kinetics400 | 95.9% | 73.5% |
> | AFNet-TSM | Mini-Kinetics | 91.5% | 65.3% |
>
> Our performance does not achieve the best as we only have the pretrained model on Mini-Kinetics which only have half the class compared with kinetics400. TSM has clear advantages over the other two methods, and our model can outperform TSM on accuracy and efficiency on Something-Something. Therefore, we believe our method can potentially get much better results if AFNet can be pretrained on Kinetics400. Nevertheless, we want to stress again that the comparison with other static methods which is designed for better temporal modeling is not a common practice in this area.
>
>
> We hope the explanations can address your concerns and we would appreciate it a lot if you can recognize the contributions of our work.

---

> ### Author Response · Authors · 2022-08-02
> **Response to Reviewer aYAw_Part 3 (reference)**
>
> [1] Feichtenhofer C, Fan H, Malik J, et al. Slowfast networks for video recognition[C]//Proceedings of the IEEE/CVF international conference on computer vision. 2019: 6202-6211.
> [2] Meng Y, Lin C C, Panda R, et al. Ar-net: Adaptive frame resolution for efficient action recognition[C]//European Conference on Computer Vision. Springer, Cham, 2020: 86-104.
> [3] Wang Y, Chen Z, Jiang H, et al. Adaptive focus for efficient video recognition[C]//Proceedings of the IEEE/CVF International Conference on Computer Vision. 2021: 16249-16258.
> [4] Sun X, Panda R, Chen C F R, et al. Dynamic network quantization for efficient video inference[C]//Proceedings of the IEEE/CVF International Conference on Computer Vision. 2021: 7375-7385.
> [5] Kondratyuk D, Yuan L, Li Y, et al. Movinets: Mobile video networks for efficient video recognition[C]//Proceedings of the IEEE/CVF Conference on Computer Vision and Pattern Recognition. 2021: 16020-16030.
> [6] He K, Zhang X, Ren S, et al. Deep residual learning for image recognition[C]//Proceedings of the IEEE conference on computer vision and pattern recognition. 2016: 770-778.
> [7] Wang L, Xiong Y, Wang Z, et al. Temporal segment networks: Towards good practices for deep action recognition[C]//European conference on computer vision. Springer, Cham, 2016: 20-36.
> [8] Lin J, Gan C, Han S. Tsm: Temporal shift module for efficient video understanding[C]//Proceedings of the IEEE/CVF International Conference on Computer Vision. 2019: 7083-7093.
> [9] Howard A, Sandler M, Chu G, et al. Searching for mobilenetv3[C]//Proceedings of the IEEE/CVF international conference on computer vision. 2019: 1314-1324.
> [10] Feichtenhofer C, Pinz A, Zisserman A. Convolutional two-stream network fusion for video action recognition[C]//Proceedings of the IEEE conference on computer vision and pattern recognition. 2016: 1933-1941.
> [11] Liu K, Liu W, Gan C, et al. T-C3D: Temporal convolutional 3D network for real-time action recognition[C]//Proceedings of the AAAI conference on artificial intelligence. 2018, 32(1).
> [12] Korbar B, Tran D, Torresani L. Scsampler: Sampling salient clips from video for efficient action recognition[C]//Proceedings of the IEEE/CVF International Conference on Computer Vision. 2019: 6232-6242.

---

> ### Author Response · Authors · 2022-08-06
> **Update of Results on MobileNet V3**
>
> Dear Reviewer aYAw,
>
> Thank you so much for helping improve our paper so far! We have an update of our results on MobileNet V3[1].
>
> ***
> **Update of Results on MobileNet V3**
> As we promised in Q5, we will build AFNet on MobileNet V3 (MN3) to test the generalization of our method on efficient backbones. Moreover, we have implemented AFNet on stronger backbone ResNet-101 (R101)[2] and we conclude both results on Something-Something V1 in this table:
> | Method  | Frame | Top-1 Acc. |  GFLOPs |
> | :-----: | ------------ | :-----: | :-----: |
> | TSM_R101 | 8 | 47.2% | 62.8G |
> | AFNet-TSM_R101(RT=0.4) | 8 | 47.2% | 28.0G |
> | TSM_R101 | 12 | 49.1% | 94.2G |
> | AFNet-TSM_R101(RT=0.4) | 12 | 49.8% | 42.1G |
> | |
> | TSM_MN3 | 8 | 42.2% | 1.7G |
> | AFNet-TSM_MN3(RT=0.4) | 8 | 43.6% | 1.5G |
> | TSM_MN3 | 12 | 43.9% | 2.6G |
> | AFNet-TSM_MN3(RT=0.4) | 12 | 45.3% | 2.2G |
>
> From this table, we can observe that AFNet continuously improves the accuracy of baseline methods while costing less computation. Specifically, when implementing on stronger backbone ResNet-101, AFNet significantly improves the efficiency by only costing 44.6% and 44.7% of the computations. When we build AFNet on efficient structure MobileNetv3, the saved computations are less obvious as the architecture of the base model is already extremely lightweight. Nevertheless, the improvement in accuracy is more obvious compared to results on ResNet-101 which can be attributed to the effectiveness of the two-branch design and navigation module.
>
> [1] Howard A, Sandler M, Chu G, et al. Searching for mobilenetv3[C]//Proceedings of the IEEE/CVF international conference on computer vision. 2019: 1314-1324.
> [2] He K, Zhang X, Ren S, et al. Deep residual learning for image recognition[C]//Proceedings of the IEEE conference on computer vision and pattern recognition. 2016: 770-778.
> ***
>
>
> Given the NeurIPS final discussion deadline (08/09) is approaching, we really hope to have a further discussion with you to see if our responses solve your concerns.
>
> Thank you so much for being with us so far! Have a wonderful weekend!
>
> Sincerely,
> Authors

---

> ### Author Response · Authors · 2022-08-09
> **Gentle reminder for the deadline**
>
> Dear Reviewer aYAw:
>
> Thank you so much for delivering your valuable comments to us. We have tried our best to provide a detailed response to you and revised the draft accordingly based on your concerns.
>
> We would like to kindly remind you that tomorrow (Aug 9th) is the deadline for the discussion phase and your opinions regarding our response matter a lot to us.
>
> Thank you so much for helping us improve our paper.
>
> Sincerely,
> Authors

---

> ### Comment · Reviewer_aYAw · 2022-08-09
> **Thank you for your response**
>
> I appreciate that the authors provide a detailed response to my concerns. They conducted additional comparisons with MobileNet v3 and more frames, which verify the effectiveness and efficiency of the proposed method. As for the novelty compared with SlowFast and Scsampler, I still have my concerns despite the differences in static/dynamic schemes and the frame/feature selections between them. I keep my rating as Borderline accept.

---

> > ### Author Response · Authors · 2022-08-09
> > **response to similarity with other works**
> >
> > Dear Reviewer aYAw,
> >
> > We thank you so much for updating the feedback to us.
> >
> > Based on your response, we want to kindly clarify several points in case there is a misunderstanding of our work and we hope these can resolve your concerns.
> >
> > ***
> > (1) The main reason for the two-branch design in AFNet is to prevent the information loss which we have observed in previous dynamic methods:
> > | Method  | Frame | mAP |
> > | :-----: | ------------ | :-----: |
> > | TSN | 16 | 76.9% |
> > | SCSampler | 16 | 72.9% |
> > | AR-Net | 16 | 73.8% |
> > | VideoIQ | 16 | 74.8% |
> > | AdaFocus | 16 | 75.0% |
> > | AFNet | 16 | 76.6% |
> >
> > When sampling 16 frames on ActivityNet, SCSampler[1], AR-Net[2], VideoIQ[3], and AdaFocus[4] have worse performance compared to TSN[5] as they completely abandon the information that the policy network recognizes as unimportant. However, AFNet exhibits similar performance (slightly inferior) compared with TSN as the two-branch design can effectively avoid the information loss caused by frame selection.
> >
> > Based on these analyses, we design the two-branch design for AFNet to prevent the loss of information. Therefore, while this structure seems similar to SlowFast[6], we believe the motivation and specific designs are different significantly.
> >
> > (2) As for SCSampler, it adopts a policy network to help select salient frames which explicitly reduces redundancy in data. Then the newly formed data will be sent to the deep network for classification. This procedure has been adopted by many dynamic methods like AR-Net, VideoIQ, and AdaFocus which we have shown in Figure 1. However, based on previous analysis, this will lead to information loss as the unimportant frames will be completely abandoned and cannot be utilized by the deep network for classification. Moreover, the policy network will introduce extra computations and complicate the training strategies as these methods have to split the training into several stages.
> >
> > In contrast, we design an extremely lightweight navigation module (19M FLOPs in a 12-frame network) which can be incorporated into the backbone network and make frame selection at intermediate features. Combining this design with our two-branch structure, we can not only prevent the information loss on focal branch, but enforce implicit temporal modeling (demonstrated in Section 3.2 and Table 1) as well. Moreover, the intermediate frame selection enables our method to be trained in an end-to-end fashion which makes it easy for implementation.
> >
> > [1] Korbar B, Tran D, Torresani L. Scsampler: Sampling salient clips from video for efficient action recognition[C]//Proceedings of the IEEE/CVF International Conference on Computer Vision. 2019: 6232-6242.
> > [2] Meng Y, Lin C C, Panda R, et al. Ar-net: Adaptive frame resolution for efficient action recognition[C]//European Conference on Computer Vision. Springer, Cham, 2020: 86-104.
> > [3] Sun X, Panda R, Chen C F R, et al. Dynamic network quantization for efficient video inference[C]//Proceedings of the IEEE/CVF International Conference on Computer Vision. 2021: 7375-7385.
> > [4] Wang Y, Chen Z, Jiang H, et al. Adaptive focus for efficient video recognition[C]//Proceedings of the IEEE/CVF International Conference on Computer Vision. 2021: 16249-16258.
> > [5] Wang L, Xiong Y, Wang Z, et al. Temporal segment networks: Towards good practices for deep action recognition[C]//European conference on computer vision. Springer, Cham, 2016: 20-36.
> > [6] Feichtenhofer C, Fan H, Malik J, et al. Slowfast networks for video recognition[C]//Proceedings of the IEEE/CVF international conference on computer vision. 2019: 6202-6211.
> > ***
> >
> > We appreciate it a lot for your further feedback and we will continuously revise the final version to highlight our contributions. Thanks again for your efforts.
> >
> > Sincerely,
> > Authors

---

### Official Review · Reviewer_ChDo · 2022-07-12

**Rating:** 6
**Confidence:** 3
**Soundness:** 2 fair
**Presentation:** 3 good
**Contribution:** 3 good

**Summary:**

This paper mainly targets action recognition. The authors advocate the importance of utilizing as many frames from each video as possible, while preserving proper computational cost. To this end they propose a novel architecture with two branches, one of which handles the whole video snippet with lower resolution and samples informative frames for fine-grained inference in another branch.

**Questions:**

1. In fact the claim that information loss leads to inferior performance is kind of counterfactual. Many previous studies have shown that for simple actions even one frame is enough for inference. On the other hand, for those complicated actions, temporal redundancy still exists. If the authors want to show the merit of utilizing all frames, it may be a good choice to directly train a model with all frames regardless of computational cost to show if it can significantly outperform the existing methods.
2. The authors mention that outputs from two branches are merged using ‘specially designed fusion strategy’ (L120). However, the fusion strategy as in Eq. 8 is simply a weighted average with learnable weight. I am afraid that the authors overclaim their contribution on this module.
3. The authors utilize residual connection between layers in the focal branch. This is questionable since different layers inquires potentially different frames due to the designed navigation module, which means features from different layers are not aligned in temporal dimension. I wonder if it is proper to directly add these features together.
4. The authors claim that the proposed method do not build temporal modeling module. I am not sure whether the $W_2$ in Eq. 2 plays such a role to interact among frames.


**Limitations:**

Several points of limitations are mentioned, which is comprehensive and provides promising future works for the current method.

**Strengths And Weaknesses:**

Strength:
1. The idea of end-to-end frame selection is interesting.
2. The authors provide abundant experiment to verify the effectiveness of their method.


Weakness:
1. The description is not clear enough. For example, the authors mention that the motivation of their method is to avoid ‘the loss of information compared to other dynamic methods’. However, it is not shown in this paper what kind of information loss exists in previous methods, and how such loss does harm to the performance. It would be better if more pilot study can be provided.
2. The proposed two-branch structure with different temporal scale is similar to SlowFast [1], which is one of the most famous action recognition models and also builds lateral connection between a slow branch and a fast branch with difference frame rates. The authors should consider discussing about this paper when introducing the proposed method.

[1] Feichtenhofer C, Fan H, Malik J, et al. Slowfast networks for video recognition[C]//Proceedings of the IEEE/CVF international conference on computer vision. 2019: 6202-6211.

---

> ### Author Response · Authors · 2022-08-02
> **Response to Reviewer ChDo_Part 1**
>
> We appreciate the Reviewer's feedback. We make further explanations to clarify the Reviewer's concerns based on several key points as below.
>
> **Weakness 1: Description not clear enough.**
> As shown in Figure 1, many existing dynamic networks (e.g., AR-Net[1], AdaFocus[2], VideoIQ[3]) adopt a policy network to select salient regions, frames, or proper resolutions for each frame which explicitly reduces redundancy in data. Then the newly formed data will be sent to the deep network for classification.
> For example, when sampling 16 frames from the original video, static methods like TSN[4] will directly use the 16 frames as the input of the deep network. While most dynamic approaches will first preprocess the 16 frames (e.g., selecting salient frames or using smaller resolutions for unimportant frames) and then send the new formed data into the deep network.
> In this manner, lots of computations will be saved. Though, there will be information lost during the preprocessing phase as the unimportant frames or regions are completely abandoned. This phenomenon motivates us to design AFNet which adopts a two-branch architecture to avoid the information loss.
> We conduct a pilot study on ActivityNet to compare multiple dynamic approaches with the baseline network TSN in the table below. Note that all methods use ResNet-50 as the backbone for classification and do not involve any temporal modeling module.
> | Method  | Frame | mAP |
> | :-----: | ------------ | :-----: |
> | TSN | 16 | 76.9% |
> | AR-Net | 16 | 73.8% |
> | VideoIQ | 16 | 74.8% |
> | AdaFocus | 16 | 75.0% |
> | AFNet | 16 | 76.6% |
> | |
> | TSN | 32 | 78.0% |
> | AFNet | 32 | 77.6% |
>
> When sampling 16 frames, AR-Net, VideoIQ, and AdaFocus have worse performance compared to TSN as they use smaller resolutions or small crops of the original data.
> However, AFNet exhibits similar performance (slightly inferior) compared with TSN as the two-branch design can effectively prevent the information loss caused by frame selection. Besides, it is worth noting that our computations are much smaller compared with TSN. Furthermore, we conduct experiments on 32 frames and the phenomenon is similar to 16 frames.
>
> **W2: Discussion with SlowFast[5].**
> We thank the Reviewer for pointing it out. However, the differences between the two methods are: (1) network category: SlowFast[5] is a static 3D model, but AFNet is a dynamic 2D network;
> (2) motivation: SlowFast is designed to capture semantic information and changing motion with branches at different temporal speed for better performance, while AFNet is aimed to dynamically skip frames to save computation and the design of two-branch structure is to prevent the information loss;
> (3) specific design: AFNet is designed to downsample features for efficiency at ample branch while SlowFast processes features in the original resolution;
> (4) temporal modeling: SlowFast applies 3D convolutions for temporal modeling, AFNet is a 2D model which is enforced with implicit temporal modeling by the designed navigation module.
> We will add this part in our final version.
>
> **Question 1: Information loss leads to inferior performance.**
> As we have explained in the reply to Weakness 1, information loss is used to describe the preprocessing phase of dynamic networks, instead of the sampling procedure for videos. From the table in that reply, we can see that existing dynamic works overlook the problem of information loss which leads to lower accuracy compared with TSN when sampling the same number of frames. Therefore, it motivates us to design the two-branch structure to avoid the information loss and achieve a better trade-off between accuracy and efficiency.
>
> **Q2: Fusion strategy.**
> We admit that there is an unclear description of this fusion strategy. Indeed, the learnable weight has been well explored in previous work [6],[7] and we tailored this design into our two-branch network for feature aggregation.
> However, we want to stress that the fusion strategy is not the main contribution of this paper. Besides, we have conducted experiments on ActivityNet to prove the effectiveness of this design:
> | Method  | mAP |
> | :-----: | ------------ |
> | AFNet(RT=0.5) | 74.3% |
> | AFNet(RT=0.5)_wo_fusion | 73.5% |
>
> The results can demonstrate that this design is non-trivial as it effectively balances the weights between the features from two branches.

---

> > ### Comment · Reviewer_ChDo · 2022-08-07
> > **Thank you for your response**
> >
> > Thank the authors for the detailed response. I still have a comment about the description of information loss. As the authors mention that the information loss is resulted from the fact that 'unimportant frames or regions are completely abandoned', which is reasonable. However, I am not sure if it is necessary to utilize dense frames as adopted in this paper to enhance the model since these unimportant frames are generally redundant and can hardly provide useful information for the target task. Therefore I asked in my former review that it would be better to have visualization or a good measure of what kind of information is contained in these unimportant frames. Another good experiment is to verify to what extent using randomly sampled unimportant frames instead of all frames can worsen the performance of the current method.

---

> > > ### Author Response · Authors · 2022-08-07
> > > **response to information loss**
> > >
> > > We appreciate a lot for the Reviewer's further feedback. Here are further explanations for the information loss.
> > >
> > > ***
> > > ***If it is necessary to utilize dense frames***
> > > A straightforward ablation is to remove the two-branch design and only use a single branch to conduct intermediate frame selection. The result is shown in Table 5: the single branch obtains much worse results than the two-branch design, especially when the selection ratio is small. This demonstrates that the two-branch structure is necessary as completely abandoning the information at ample branch will lead to much worse results.
> > >
> > > ***Visualization of what kind of information is contained in unimportant frames***
> > > Thanks for this question. However, it is hard to analyze the information in unimportant frames as different frames will be selected at different layers. Therefore, we analyze the learned policy by calculating the distributions of RT at different stages.  In Figure 5, we can observe a decreased trend in RT for all curves which means that AFNet tends to select more frames at earlier layers and skip more at later stages. It can be explained that earlier layers mostly capture low-level information which has relatively large divergence among different frames. While high-level semantics between different frames are more similar, therefore AFNet tends to skip more at later convolution blocks.
> > >
> > > ***Using randomly sampled unimportant frames instead of all frames***
> > > If we understand the Reviewer's comment correctly, we are suggested to use the unimportant frames on ample branch instead of all frames. In our original implementation, we will learn binary temporal masks $L_n$ to select important frames. Therefore, we can generate complementary masks $L_n'$ to choose the unimportant frames on ample branch. The results are shown in table:
> > > | Method  | Frame | mAP | GFLOPs |
> > > | :-----: | ------------ | :-----: | :-----: |
> > > | AFNet(RT=0.5)_all_frames | 12 | 74.3% | 29.7G |
> > > | AFNet(RT=0.5)_unimportant_frames | 12 | 72.3% | 28.4G |
> > >
> > > We can see that using unimportant frames on ample branch clearly lead to worse performance in accuracy. This demonstrates the necessity to keep all frames on the ample branch. Moreover, utilizing all the frames on ample branch does not cause too much difference in computational costs (only 1.3G for 12 frames). The reason is that the ample branch is designed to be very lightweight as it will downsample the features and reduce the channel size. Therefore, the two-branch design is effective as it prevents the information loss while only costs small computations.
> > >
> > > Furthermore, we have demonstrated in Table 5 that our navigation module is also effective as it exhibits better performance compared to other selection strategies.
> > > ***
> > >
> > > Based on these points above, we think it is necessary to utilize all frames on the ample branch to prevent the information loss.
> > >
> > > Thank you so much for delivering further feedback to us. If you have further comments or concerns, feel free to ask and we are willing to have further discussion with you. Thanks again.

---

> ### Author Response · Authors · 2022-08-02
> **Response to Reviewer ChDo_Part 2**
>
>
> **Q3: Misaligning in temporal dimension.**
> Thanks for asking this question. Indeed, the navigation module will select different frames at different convolutional blocks. During training, the unselected frames will be masked by zero values so that the features will have the same shape and we can directly add them to the residual feature.
> To avoid computation on the unimportant frames during inference, we will extract the salient frames from the original feature (slice operation on temporal dimension) before sending it to the convolutions. After that, we will create a zero tensor and rearrange the processed frames to the original temporal locations based on the learned mask.
> In this manner, we can prevent the misaligning during training and achieve efficiency during inference.
>
> **Q4: Equation 3 plays a role in temporal modeling.**
> Yes, $W_2$ in Equation (3) models the temporal relations between frames by transforming the temporal order to channel dimension. However, this logit is used to generate a binary temporal mask for frame selection, instead of used for temporal modeling. Even though, we have demonstrated in Section 3.2 that the binary mask results in learnable temporal weights which implicitly enforce temporal modeling. The results in Table 1 also prove this point as AFNet outperforms TSN by 9.1% due to implicit temporal modeling.
>
>
> We hope the explanations can address your concerns and we would appreciate it a lot if you can recognize the contributions of our work.
>
>
> [1] Meng Y, Lin C C, Panda R, et al. Ar-net: Adaptive frame resolution for efficient action recognition[C]//European Conference on Computer Vision. Springer, Cham, 2020: 86-104.
> [2] Wang Y, Chen Z, Jiang H, et al. Adaptive focus for efficient video recognition[C]//Proceedings of the IEEE/CVF International Conference on Computer Vision. 2021: 16249-16258.
> [3] Sun X, Panda R, Chen C F R, et al. Dynamic network quantization for efficient video inference[C]//Proceedings of the IEEE/CVF International Conference on Computer Vision. 2021: 7375-7385.
> [4] Wang L, Xiong Y, Wang Z, et al. Temporal segment networks: Towards good practices for deep action recognition[C]//European conference on computer vision. Springer, Cham, 2016: 20-36.
> [5] Feichtenhofer C, Fan H, Malik J, et al. Slowfast networks for video recognition[C]//Proceedings of the IEEE/CVF international conference on computer vision. 2019: 6202-6211.
> [6] Hu J, Shen L, Sun G. Squeeze-and-excitation networks[C]//Proceedings of the IEEE conference on computer vision and pattern recognition. 2018: 7132-7141.
> [7] Li X, Wang W, Hu X, et al. Selective kernel networks[C]//Proceedings of the IEEE/CVF conference on computer vision and pattern recognition. 2019: 510-519.

---

> ### Author Response · Authors · 2022-08-06
> **Sincerely looking forward to your further feedback!**
>
> Dear Reviewer ChDo,
>
> Thank you so much for helping improve our paper so far!
>
> Given the NeurIPS final discussion deadline (08/09) is approaching, we really hope to have a further discussion with you to see if our responses solve your concerns.
>
> Thank you so much for being with us so far! Have a wonderful weekend!
>
> Sincerely,
> Authors

---

### Official Review · Reviewer_VKUT · 2022-07-15

**Rating:** 6
**Confidence:** 5
**Soundness:** 3 good
**Presentation:** 3 good
**Contribution:** 3 good

**Summary:**

This paper proposes a two-branch network for efficient video recognition. In particular, the Ample Branch takes densely sampled input frames and processes them with reduced channel sizes. On the other hand, the Focal Branch only processes salient frames selected by a navigation module. Extensive experiments on multiple video benchmarks show that the proposed AFNet achieves state-of-the-art results with lower computational cost.

**Questions:**

1. The GFLOPs of RT=0.8 is a bit counterintuitive. Considering (1) the introduction of the addition Amber Branch, navigation module and (2) the last stage of the network remains unchanged, the GFLOPs of the model with RT=0.8 should be larger than 80% of that of the backbone model. This trend can be observed for the setting RT=0.4. However, for RT=0.8, GFLOPs = 31.7 is only 64.6% of TSM (GFLOPs=49.1), which is just slightly larger than RT=0.4 (GFLOPs=27.8). Please clarify this in the rebuttal.

2. For the navigation module, instead of learning a sampling strategy, what if we learn a linear function that transforms the original T frames to T' steps? In other words, we aim to learn a T x T' weight matrix that compute T' weighted averages of the T frames. It's differentiable and easy to optimized with the rest of the model, and I'm curious whether it'll give worse results than doing frame sampling.

**Limitations:**

Yes.

**Strengths And Weaknesses:**

Strength
1. The proposed architecture, AFNet, is simple yet effective for efficient video action recognition. As shown in the experiment section, AFNet achieves even better results than its baseline with lower computational cost. The idea of leveraging more input frames for avoiding information loss and salient frame selection is well-motivated, and the encouraging results provided in this paper are potentially insightful for the research community.
2. The paper is overall well organized and well written.

Weakness
1. The novelty of individual components of AFNet is limited. For example, the two-branch design with downsampled "fast branch" is explored in SlowFast network (without dynamic selection of salient frames though);  the navigation module along with the Gumbel softmax optimization technique is also used in prior work for dynamic selection [1]. However, I believe that the contribution of the overall architecture design is sufficient and the proposed method achieves good results.

2. Since Gumbel softmax is used for dynamic sampling of frames, the computational cost cannot be reduced during training. Although we usually care more about the computational cost of a model at inference, the cost of training will become a bottleneck if (1) the model is too large to fit into GPU memory (e.g., using more frames than 12 frames used in the paper); (2) the model training takes too much time. I understand it's out of the scope of this paper, but I'd suggest the author to try other "hard-sampling" algorithms for the selection module, for example, the perturbed maximum method [2, 3].

3. Because the proposed two-branch design is generic to different backbone models, it's always better to see more experimental results with stronger backbones (e.g., ResNet-101 or even Transformers) and using more frames (12 frames are still a small number for long videos such as those in ActivityNet).


[1] Rao, Y., Zhao, W., Liu, B., Lu, J., Zhou, J., Hsieh, C.J.: Dynamicvit: Efficient vision transformers with dynamic token sparsification. In: NIPS (2021)
[2] Berthet, Quentin, et al. "Learning with differentiable pertubed optimizers." Advances in neural information processing systems 33 (2020): 9508-9519.
[3] Cordonnier, Jean-Baptiste, et al. "Differentiable patch selection for image recognition." Proceedings of the IEEE/CVF Conference on Computer Vision and Pattern Recognition. 2021.

---

> ### Author Response · Authors · 2022-08-02
> **Response to Reviewer VKUT_Part 1**
>
> We appreciate the Reviewer’s approval and constructive suggestions for us to improve our work. We make the response as below.
>
> **Weakness 1: Relation to SlowFast and Dynamic Selection.**
> ***SlowFast***: Good point. Yet, the differences are: (1) network category: SlowFast[1] is a static 3D model, but AFNet is a dynamic 2D network;
> (2) motivation: SlowFast is designed to capture semantic information and changing motion with branches at different temporal speed for better performance, while AFNet is aimed to dynamically skip frames to save computation and the design of two-branch structure is to prevent the information loss;
> (3) specific design: AFNet is designed to downsample features for efficiency at ample branch while SlowFast processes features in the original resolution;
> (4) temporal modeling: SlowFast applies 3D convolutions for temporal modeling, AFNet is a 2D model which is enforced with implicit temporal modeling by the designed navigation module.
>
> ***Dynamic Selection***: We admit that Gumbel softmax optimization[2] has been well explored in previous works. However, employing Gumbel softmax optimization at intermediate features for frame selection has been seldom touched in this area.
> The design of combining navigation module with the two-branch structure is nontrivial as we demonstrated it enforces implicit temporal modeling in Section 3.2 and the experiments in Table 1. Concretely, AFNet outperforms TSN[3] by 9.1% on Something-Something V1 with the help of implicit temporal modeling.
>
> We really appreciate it that the Reviewer recognizes our contributions and we will add this discussion in our final version.
>
> **Weakness2: Hard-sampling methods.**
> Thanks for the great suggestion.
> We leave the perturbed maximum method in our future works due to limited time during rebuttal.
> However, we would like to emphasize that one advantage of AFNet over many other dynamic networks is that we can train the network in an end-to-end fashion without extending the training epochs.
> However, previous works like AR-Net[4], VideoIQ[5] have to split the training into multiple stages as they introduce an extra policy network to make decisions.
>
> **Weakness3: Stronger backbones and more frames.**
> First, we conduct experiments on Somethin-Something V1 for backbone based on ResNet-101[6].
> | Method  | Frame | Top-1 Acc. |  GFLOPs |
> | :-----: | ------------ | :-----: | :-----: |
> | TSM_101 | 8 | 47.2% | 62.8G |
> | AFNet-TSM_101(RT=0.4) | 8 | 47.2% | 28.0G |
> | TSM_101 | 12 | 49.1% | 94.2G |
> | AFNet-TSM_101(RT=0.4) | 12 | 49.8% | 42.1G |
>
> The results show that our method generalizes well to stronger backbones. Then we conduct experiments on ActivityNet with more sampled frames:
> | Method  | Frame | mAP |  GFLOPs |
> | :-----: | ------------ | :-----: | :-----: |
> | TSN | 16 | 76.9% | 65.6G |
> | AFNet(RS=0.4,RT=0.8) | 16 | 76.6% | 32.9G |
> | TSN | 32 | 78.0% | 131.2G |
> | AFNet(RS=0.4,RT=0.8) | 32 | 77.6% | 60.9G |
>
> When sampling more frames, AFNet costs significantly less computations compared to baseline method with only a slight drop in performance which can be attributed to two-branch structure as it avoids the information loss (more analysis can be found in our reply to weakness 1 pointed out by Reviewer ChDo). Thanks for the advice and we will add this part in our revision.
>
> **Question 1: FLOPs counterintuitive.**
> ***GFLOPs of RT=0.8 is only 64.6% of TSM:*** Thanks for carefully reading our paper.
> The reason is due to the two-branch structure: (1) the ample branch is designed to squeeze the spatial and channel size by a factor of two which makes this branch very lightweight. In this way, we can prevent the loss of information by keeping all the frames at ample branch but with minimal costs.
> (2) we only compute the salient frames at the focal branch and the group number of convolutions at this branch is set to two to further reduce the cost (illustrated in lines 153,154).
> (3) our navigation module is designed to be very lightweight as the computational cost is only 19M in this RT=0.8 model. Based on these reasons above, the GFLOPs of AFNet is very small.
>
> ***GFLOPs of RT=0.8 is slight larger than RT=0.4:*** This is because we only conduct frame selection at focal branch and the saved computations can only be obtained through this branch which results in a non-linear deduction in GFLOPs. Though, AFNet still achieves a smaller cost compared to other methods owing to the design of our two-branch structure and dynamic selection of navigation module.

---

> ### Author Response · Authors · 2022-08-02
> **Response to Reviewer VKUT_Part 2**
>
>
> **Q2: Learning weight matrix.**
> If we understand the Reviewer's comment correctly, we are suggested to learn a soft weight for each frame.
> We conduct the experiment by removing gumbel softmax in our navigation module and modifying it to learn a soft temporal attention for the T frames in focal branch. The result is shown below.
>
> | Method  | Top-1 Acc. |
> | :-----: | ------------ |
> | TSN | 18.6% |
> | AFNet(RT=1.00) | 19.2% |
> | AFNet(RT=0.50) | 26.8% |
> | AFNet(RT=0.25) | 27.7% |
> | AFNet(soft_weight) | 27.0% |
>
> We can make several conclusions from the table:
> (1)	the learned weights significantly improve the performance of AFNet(RT=1.00) as it did not build any temporal modeling module like TSN, and the gain in performance can again demonstrate the effectiveness of our navigation module.
> (2)	AFNet(soft_weight) has a similar performance to AFNet(RT=0.25) and AFNet(RT=0.5) which meets our expectations as we have analyzed in Section 3.2.
> Though the navigation module just learns a binary mask for each frame, it will decide whether the coefficient will be calculated for each frame at every convolutional block which results in learned temporal weights in each video. Learning a soft weight cause the same effect.
> Thanks for this great suggestion, we will add this part in our revision as it can help to better explain our implicit temporal modeling.
>
> [1] Feichtenhofer C, Fan H, Malik J, et al. Slowfast networks for video recognition[C]//Proceedings of the IEEE/CVF international conference on computer vision. 2019: 6202-6211.
> [2] Jang E, Gu S, Poole B. Categorical reparameterization with gumbel-softmax[J]. arXiv preprint arXiv:1611.01144, 2016.
> [3] Wang L, Xiong Y, Wang Z, et al. Temporal segment networks: Towards good practices for deep action recognition[C]//European conference on computer vision. Springer, Cham, 2016: 20-36.
> [4] Meng Y, Lin C C, Panda R, et al. Ar-net: Adaptive frame resolution for efficient action recognition[C]//European Conference on Computer Vision. Springer, Cham, 2020: 86-104.
> [5] Sun X, Panda R, Chen C F R, et al. Dynamic network quantization for efficient video inference[C]//Proceedings of the IEEE/CVF International Conference on Computer Vision. 2021: 7375-7385.
> [6] He K, Zhang X, Ren S, et al. Deep residual learning for image recognition[C]//Proceedings of the IEEE conference on computer vision and pattern recognition. 2016: 770-778.

---

> ### Author Response · Authors · 2022-08-07
> **Sincerely looking forward to your further feedback!**
>
> Dear Reviewer VKUT,
>
> Thank you so much for helping improve our paper so far! We have an update of our results on MobileNet V3[1].
>
> ***
> **Update of Results on MobileNet V3**
> To better demonstrate the generalization of AFNet, we build AFNet on MobileNet V3 (MN3) to test its performance on efficient backbones. We conclude both results of ResNet-101[2] and MobileNet V3 on Something-Something V1 in this table:
> | Method  | Frame | Top-1 Acc. |  GFLOPs |
> | :-----: | ------------ | :-----: | :-----: |
> | TSM_R101 | 8 | 47.2% | 62.8G |
> | AFNet-TSM_R101(RT=0.4) | 8 | 47.2% | 28.0G |
> | TSM_R101 | 12 | 49.1% | 94.2G |
> | AFNet-TSM_R101(RT=0.4) | 12 | 49.8% | 42.1G |
> | |
> | TSM_MN3 | 8 | 42.2% | 1.7G |
> | AFNet-TSM_MN3(RT=0.4) | 8 | 43.6% | 1.5G |
> | TSM_MN3 | 12 | 43.9% | 2.6G |
> | AFNet-TSM_MN3(RT=0.4) | 12 | 45.3% | 2.2G |
>
> From this table, we can observe that AFNet continuously improves the accuracy of baseline methods while costing less computation. Specifically, when implementing on stronger backbone ResNet-101, AFNet significantly improves the efficiency by only costing 44.6% and 44.7% of the computations. When we build AFNet on efficient structure MobileNetv3, the saved computations are less obvious as the architecture of the base model is already extremely lightweight. Nevertheless, the improvement in accuracy is more obvious compared to results on ResNet-101 which can be attributed to the effectiveness of the two-branch design and navigation module.
>
> [1] Howard A, Sandler M, Chu G, et al. Searching for mobilenetv3[C]//Proceedings of the IEEE/CVF international conference on computer vision. 2019: 1314-1324.
> [2] He K, Zhang X, Ren S, et al. Deep residual learning for image recognition[C]//Proceedings of the IEEE conference on computer vision and pattern recognition. 2016: 770-778.
> ***
>
>
> Given the NeurIPS final discussion deadline (08/09) is approaching, we really hope to have a further discussion with you to see if our responses solve your concerns.
>
> Thank you so much for being with us so far!
>
> Sincerely,
> Authors

---

> ### Author Response · Authors · 2022-08-09
> **Gentle reminder for the deadline**
>
> Dear Reviewer VKUT:
>
> Thank you so much for delivering your valuable comments to us. We have tried our best to provide a detailed response to you and revised the draft accordingly based on your concerns.
>
> We would like to kindly remind you that tomorrow (Aug 9th) is the deadline for the discussion phase and your opinions regarding our response matter a lot to us.
>
> Thank you so much for helping us improve our paper.
>
> Sincerely,
> Authors

---

### Author Response · Authors · 2022-08-08
**Updated Revision**

Dear Reviewers:

Thanks for your valuable comments made in the review process. We have revised the draft based on your suggestions and the revised area is marked in blue color. Specifically, we have:
***
* added the discussion with SlowFast[1] in Section 2 and the comparison in Table 2,
* built AFNet on ResNet-101[2] and MobileNet V3[3] to prove its generalization ability in Section 3 of supplementary material,
* built AFNet with more frames on ActivityNet to demonstrate its efficiency in Section 4 of supplementary material,
* tested the practical speedup of AFNet on CPU and GPU and compared it with competing methods in Section 7 of supplementary material,
* added ablation of fusion strategy and temperature decay schedule in Section 8 of supplementary material,
* demonstrated our analysis in Section 3.2 by learning soft temporal weights for each video in Section 5 of supplementary material,
* added the description of computational paradigms for training and inference in Section 6 of supplementary material,
* revised the typos and unclear description.

[1] Feichtenhofer C, Fan H, Malik J, et al. Slowfast networks for video recognition[C]//Proceedings of the IEEE/CVF international conference on computer vision. 2019: 6202-6211.
[2] He K, Zhang X, Ren S, et al. Deep residual learning for image recognition[C]//Proceedings of the IEEE conference on computer vision and pattern recognition. 2016: 770-778.
[3] Howard A, Sandler M, Chu G, et al. Searching for mobilenetv3[C]//Proceedings of the IEEE/CVF international conference on computer vision. 2019: 1314-1324.
***

Thank you so much for being with us so far.

Sincerely,
Authors

---

### Meta-Review · Area_Chair_ckFi · 2022-08-26

**Recommendation:** Accept
**Confidence:** Less certain

**Metareview:**

The proposed architecture, AFNet, is simple yet effective for end-to-end efficient video action recognition.
It has good idea, well written paper, and relative solid experimental results to support the claim.
The emergency reviewer gives the highest score (6), while the other reviewers do have some concerns with this paper.
Most of the other reviewers give the rating of borderline accept, after the authours make more efforts in the rebuttal period. In term of these, the initial recommendation would be acceptance.

**Award:**

No

---

### Decision · Program_Chairs · 2022-09-14

Accept